# Segment Anything in High Quality

**Lei Ke**[*1,2]  **Mingqiao Ye**[*1]  **Martin Danelljan**[1]  **Yifan Liu**[1]  **Yu-Wing Tai**[3]
**Chi-Keung Tang**[2]  **Fisher Yu**[1]
[1]ETH Zürich          [2]HKUST          [3]Dartmouth College

## Abstract

The recent Segment Anything Model (SAM) represents a big leap in scaling up segmentation models, allowing for powerful zero-shot capabilities and flexible prompting. Despite being trained with 1.1 billion masks, SAM's mask prediction quality falls short in many cases, particularly when dealing with objects that have intricate structures. We propose HQ-SAM, equipping SAM with the ability to accurately segment any object, while maintaining SAM's original promptable design, efficiency, and zero-shot generalizability. Our careful design reuses and preserves the pre-trained model weights of SAM, while only introducing minimal additional parameters and computation. We design a learnable High-Quality Output Token, which is injected into SAM's mask decoder and is responsible for predicting the high-quality mask. Instead of only applying it on mask-decoder features, we first fuse them with early and final ViT features for improved mask details. To train our introduced learnable parameters, we compose a dataset of 44K fine-grained masks from several sources. HQ-SAM is only trained on the introduced detaset of 44k masks, which takes only 4 hours on 8 GPUs. We show the efficacy of HQ-SAM in a suite of 10 diverse segmentation datasets across different downstream tasks, where 8 out of them are evaluated in a zero-shot transfer protocol. Our code and pretrained models are at `https://github.com/SysCV/SAM-HQ`.

## 1  Introduction

Accurate segmentation of diverse objects is fundamental for a wide range of scene understanding applications, including image/video editing, robotic perception, and AR/VR. Trained with billion-scale mask labels, the Segment Anything Model (SAM) [21] was recently released as a foundational vision model for general image segmentation. SAM is capable of segmenting a wide range of objects, parts, and visual structures in diverse scenarios, by taking a prompt consisting of points, a bounding box, or a coarse mask as input. Its zero-shot segmentation abilities have led to a rapid paradigm shift, as it can be transferred to numerous applications through simple prompting.

While SAM has achieved impressive performance, its segmentation results are still unsatisfactory in many cases. In particular, SAM suffers from two key problems: 1) Coarse mask boundaries, often even neglecting the segmentation of thin object structures, as shown in Figure 1. 2) Incorrect predictions, broken masks, or large errors in challenging cases. This is often related to SAM misinterpreting thin structures, such as the kite lines in the rightmost column of Figure 1. These types of failures severely limit the applicability and effectiveness of foundational segmentation models, such as SAM, in particular for automated annotation and image/video editing tasks, where highly accurate image masks are crucial.

We propose HQ-SAM, which can predict highly accurate segmentation masks, even in very challenging cases (see Figure 1), without compromising the strong zero-shot capabilities and flexibility of the

---

[*]Equal contribution.

37th Conference on Neural Information Processing Systems (NeurIPS 2023).

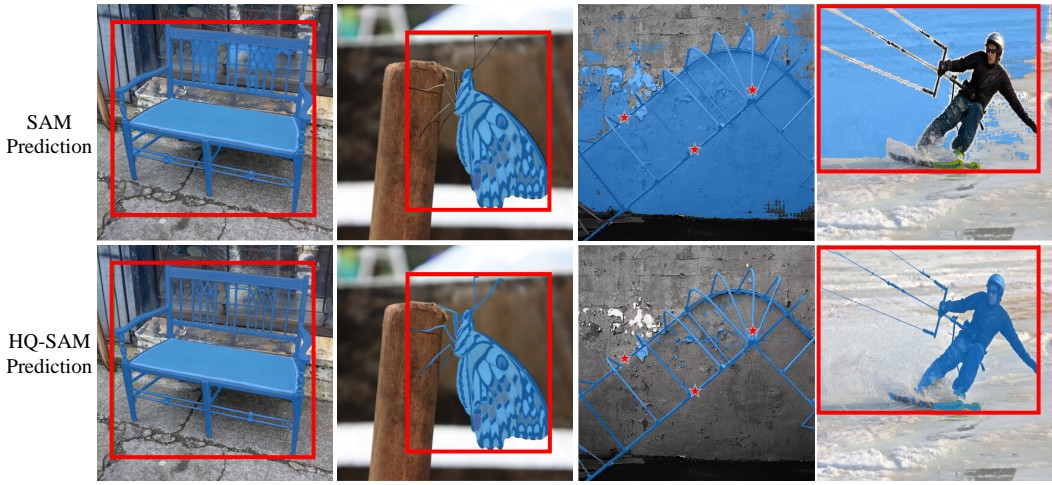

Figure 1: The predicted masks of SAM **vs.** our HQ-SAM, given the same red box or several points on the object as input prompts. HQ-SAM produces significantly more detailed results with very accurate boundaries. In the rightmost column, SAM misinterprets the thin structure of the kite lines, and produces a large portion of errors with broken holes for the input box prompt.

original SAM. To preserve the efficiency and zero-shot performance, we propose a minimal adaptation of SAM, adding less than $0.5\%$ parameters, to extend its capability to high-quality segmentation.

Directly fine-tuning the SAM decoder or introducing a new decoder module severely degrades the general zero-shot segmentation performance. We therefore propose the HQ-SAM architecture, which tightly integrates with and re-uses the existing learned SAM structure, in order to fully preserve the zero-shot performance. First, we design a learnable HQ-Output Token that is input to SAM's mask decoder, alongside the original prompt and output tokens. Unlike the original output tokens, our HQ-Output Token and its associated MLP layers are trained to predict a high-quality segmentation mask. Second, instead of only re-using the SAM's mask decoder features, our HQ-Output Token operates on a refined feature set to achieve accurate mask details. In particular, we use both global semantic context and local fine-grained features by fusing SAM's mask decoder features with early and late feature maps from its ViT encoder. During training, we freeze the entire pre-trained SAM parameters, while only updating our HQ-Output Token, its associated three-layer MLPs, and a small feature fusion block.

Learning accurate segmentation requires a dataset with accurate mask annotations of diverse objects with complex and detailed geometries. SAM is trained on the SA-1B dataset, which contains 11M images with 1.1 billion masks automatically generated by a SAM-like model. However, using this extensive dataset presents significant cost implications and falls short of achieving the desired high-quality mask generations pursued in our work, as evident by SAM's performance in Figure 1. Consequently, we compose a new dataset, called HQSeg-44K, which contains 44K extremely fine-grained image mask annotations. HQSeg-44K is constructed by merging six existing image datasets [35, 29, 26, 38, 8, 46] with highly accurate mask labels, covering over 1,000 diverse semantic classes. Thanks to the smaller-scale dataset and our minimal integrated architecture, HQ-SAM can be trained in only 4 hours on 8 RTX 3090 GPUs.

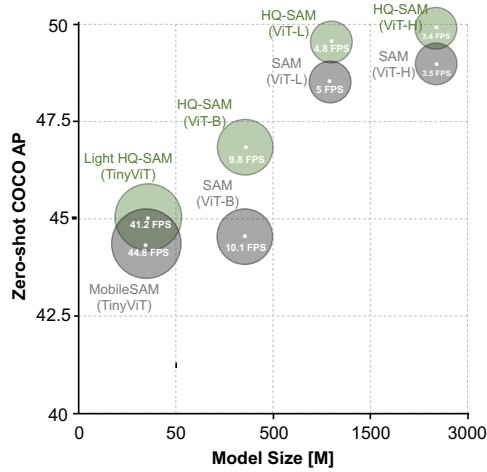

Figure 2: Performance vs. speed vs. model size for an array of SAM variants [21, 52].

To validate the effectiveness of HQ-SAM, we perform extensive quantitative and qualitative experimental analysis. We provide a comprehensive performance-speed-model size comparison on SAM variants [21, 52] in Figure 2. We compare HQ-SAM with SAM on a suite of 10 diverse segmentation datasets across different downstream tasks, where 8 out of them are under a zero-shot transfer protocol, including COCO [31], UVO [42], SGinW [58], LVIS [14], HQ-YTVIS [20], BIG [6], COIFT [29]

and HR-SOD [51]. This rigorous evaluation demonstrates that the proposed HQ-SAM can produce higher-quality masks while maintaining the zero-shot capability compared with SAM.

## 2 Related Work

**High-quality Segmentation** Existing works for high-quality segmentation are mostly trained for a specific segmentation task, like image and video instance segmentation [22, 19, 20, 40, 44], semantic segmentation [30, 54, 39, 50] or panoptic segmentation [9], in a close-world paradigm. Some of them focus on post-segmentation refinement using with graphical models such as CRF [23] or region growing [10]. However, the CRF-based refinement is adhere to low-level color boundaries without fully utilizing high-level semantic context and cannot fix large segmentation errors. While some refinement-based works adopt separate deep networks for cascade iterative refinement [6, 37], they are prone to overfitting as shown by our experiment. Compared to these high-quality segmentation [19, 22, 33] or segmentation refinement methods, we focus on accurately segmenting diverse objects on new data with flexible prompting, and build a high-quality zero-shot segmentation model that generalizes to various segmentation tasks and domains. Unlike the post segmentation refinement works [6, 37], to preserve the zero-shot segmentation capability of SAM, HQ-SAM predicts the new high-quality mask directly by reusing the image encoder and mask decoder of SAM, instead of taking the coarse mask and images as the input and feeding it into a separate refinement network. The model architecture of HQ-SAM builds upon SAM with negligible overhead, where we propose efficient token learning for accurate mask predictions. This is completely different from previous high-quality segmentation works, and we show its effectiveness across a wide range of zero-shot experiments.

**Fine-tuning and Prompt Tuning for Foundation Models** Foundation models [2, 1] first appear in the NLP community, where large language models such as GPT series [2] show strong zero-shot generalization to unseen tasks and data. Then, some prompt-based learning works [16, 27, 17] are proposed to help these pre-trained models generalize to the downstream tasks instead of fine-tuning the internal model parameters [15] for better transfer learning. For vision-based foundation models [21, 43, 59], prompt engineering [56, 45, 49, 57] that freezes the pre-trained model is first explored in vision-language models, such as CLIP [36]. These prompts with learnable parameters are designed to help downstream tasks with better context optimization. Different from the existing prompt-based or finetuning works, we focus on the minimal adaptation of SAM toward high-quality segmentation. We directly use the proposed HQ-Output Token output for accurate mask prediction, instead of only leveraging some learnable parameters [56] to help context learning and better generalization.

## 3 Method

We propose HQ-SAM to upgrade SAM for high-quality zero-shot segmentation. HQ-SAM is lightweight and only introduces two important adaptations to the SAM model. In Sec 3.1, we first briefly review the architecture of SAM on which HQ-SAM is built. Then, in Sec 3.2, we introduce our HQ-SAM with High-Quality Token (HQ-Output Token) and Global-local Feature Fusion, which are the key components to achieve better segmentation quality for SAM while preserving its zero-shot capability. Finally, in Sec 3.3, we describe the training and inference process of HQ-SAM, which is both data and computationally efficient.

### 3.1 Preliminaries: SAM

SAM [21] is composed of three modules: **(a)** Image encoder: a heavy ViT-based backbone for image feature extraction, resulting in image embedding in spatial size $64 \times 64$. **(b)** Prompt encoder: encoding the interactive positional information from the input points/boxes/masks to provide for the mask decoder. **(c)** Mask decoder: a two-layer transformer-based decoder takes both the extracted image embedding with the concatenated output and prompt tokens for final mask prediction. The released SAM model is trained on the large-scale SA-1B dataset, which contains over 1 billion automatically generated masks ($400 \times$ more masks than any existing segmentation datasets [14, 24]) and 11 million images. Thus, SAM shows valuable strong zero-shot generalization to new data without the necessity for additional training. However, we also note that SAM training is very expensive, where distributively training ViT-H-based SAM for 2 epochs on SA-1B requires 256 GPUs with a large batch size of 256 images. For more SAM method details, we refer readers to [21].

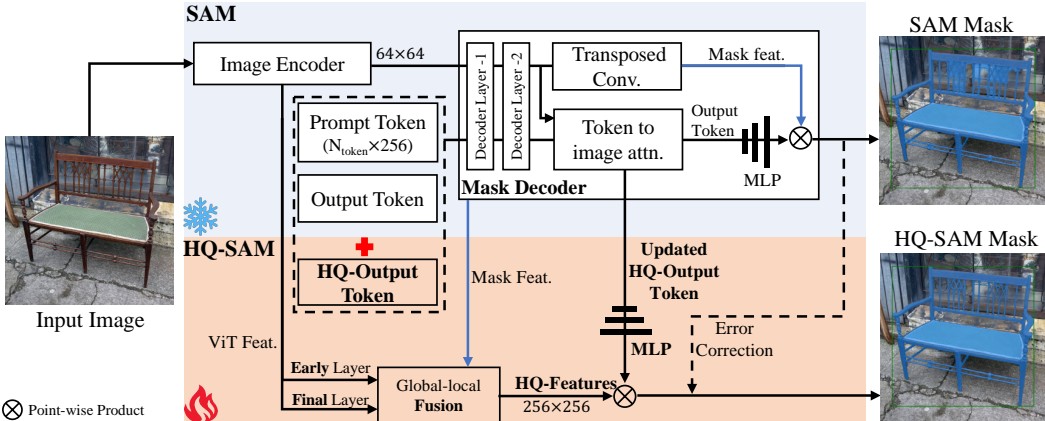

Figure 3: HQ-SAM introduces HQ-Output Token and Global-local Feature Fusion to SAM for high-quality mask prediction. To keep the zero-shot capability of SAM, the lightweight HQ-Output Token reuses SAM's mask decoder, and generates new MLP layers for performing point-wise product with fused HQ-Features. During training, only a few learnable parameters in HQ-SAM are trainable while we fix the model parameters of the pre-trained SAM. The prompt encoder is omitted here for clarity. Error correction is simply used as a direct element-wise sum between the predicted logits of the SAM's Output Token and the HQ-Output Token during inference.

## 3.2 Ours: HQ-SAM

In this section, we describe the architecture of the HQ-SAM network. To preserve the zero-shot transfer capability of SAM, while preventing model overfitting or catastrophic forgetting, instead of directly finetuning SAM or adding a new heavy decoder network, we take a minimal adaptation approach as much as possible. To this end, HQ-SAM reuses the pre-trained model weights of SAM as much as possible with only two new key components, namely, High-Quality Output Token and Global-local Feature Fusion, as illustrated in Figure 3. HQ-SAM can thus be regarded as a high-quality zero-shot segmentation model evolved from SAM with negligible extra model parameters and computation cost.

### 3.2.1 High-Quality Output Token

We propose efficient token learning for improving the mask quality of SAM. As shown in Figure 3, in SAM's original mask decoder design, the output token (similar to object query in DETR [3]) is adopted for mask prediction, which predicts dynamic MLP weights and then performs point-wise product with the mask features. To promote SAM's mask quality in HQ-SAM, instead of directly taking SAM's coarse masks as input, we introduce the HQ-Output token and a new mask prediction layer for high-quality mask prediction.

In Figure 3, by reusing and fixing SAM's mask decoder, a new learnable HQ-Output Token (size of $1 \times 256$) is concatenated with SAM's output tokens (size of $4 \times 256$) and prompt tokens (size of $N_{prompt} \times 256$) as the input to the SAM's mask decoder. Similar to the original output token, in each attention layer, HQ-Output Token first performs self-attention with other tokens and then conducts both token-to-image and the reverse image-to-token attention for its feature updating. Note that HQ-Output Token uses the point-wise MLP shared by the other tokens in each decoder layer. After passing through two decoder layers, the updated HQ-Output Token has access to the global image context, the critical geometric/type information of prompt tokens as well as hidden mask information of the other output tokens. Finally, we add a new three-layer MLP to generate dynamic convolutional kernels from the updated HQ-Output Token, which then performs spatially point-wise product with the fused HQ-feature for high-quality mask generation.

Instead of directly finetuning SAM or further adding a heavy post-refinement network, we only allow the HQ-Output Token and its associated three-layer MLPs to be trained for correcting the mask errors of SAM's output token. This is completely different from existing high-quality segmentation models [19, 6, 20, 22]. We identify two main advantages of our efficient token learning through extensive experiments: 1) This strategy significantly improves SAM's mask quality while only

introducing negligible parameters compared to original SAM, making HQ-SAM training extremely time and data-efficient; 2) The learned token and MLP layers do not overfit to mask the annotation bias of a specific dataset, thus keeping SAM's strong zero-shot segmentation capability on new images without catastrophic knowledge forgetting.

### 3.2.2 Global-local Fusion for High-quality Features

Very accurate segmentation also requires input image feature with both rich global semantic context and local boundary details. To further promote mask quality, we enrich both the high-level object context and low-level boundary/edge information in the mask decoder features of SAM. Instead of directly using SAM's mask decoder feature, we compose the new high-quality features (HQ-Features) by extracting and fusing features from different stages of the SAM model: **1)** The early layer **local** feature of SAM's ViT encoder with spatial shape $64\times64$, which captures more general image edge/boundary details [12]. Concretely, we extract the feature after the first global attention block of the ViT encoder, and for ViT-Large based SAM, this is the 6th block output for the 24 blocks in total; **2)** The final layer **global** feature of SAM's ViT encoder with shape $64\times64$, which has more global image context information; **3)** The mask feature in SAM's mask decoder with size $256\times256$, which is also shared by the output tokens, contains strong mask shape information.

As shown in Figure 3, to obtain the input HQ-Features, we first upsample the early-layer and final-layer encoder features to the spatial size $256\times256$ by transposed convolution. Then, we sum up these three types of features in an element-wise manner after simple convolutional processing. We show that this global-local feature fusion is simple while effective, yielding detail-preserving segmentation results with a small memory footprint and computation burden. We also perform detailed ablation on the effect of each feature source in the experimental section (Table 3).

### 3.3 Training and Inference of HQ-SAM

**Training Data Construction**  To train HQ-SAM in a data-efficient manner, instead of further training on SA-1B [21], we compose a new training dataset HQSeg-44K which contains 44,320 extremely accurate image mask annotations. We note that the released SA-1B dataset only contains automatically generated mask labels, missing very accurate manual annotation on objects with complex structures. Due to the annotation difficulty, HQSeg-44K leverages a collection of six existing image datasets including DIS [35] (train set), ThinObject-5K [29] (train set), FSS-1000 [26], ECSSD [38], MSRA-10K [8], DUT-OMRON [46] with extremely fine-grained mask labeling, where each of them contains 7.4K mask labels on average. To make HQ-SAM robust and generalizable to new data, HQSeg-44K contains diverse semantic classes of more than 1,000. We show the advantage of using HQSeg-44K by comparing HQ-SAM training with 44K randomly sampled images and masks from SA-1B [21] in our supplemental analysis.

**HQ-SAM Training**  During training, we fix the model parameters of the pre-trained SAM model while only making the proposed HQ-SAM learnable. The learnable parameters thus only include the HQ-Output Token, its associated three-layer MLP and three simple convolutions for HQ-Features fusion. Since SAM is designed for flexible segmentation prompts, we train HQ-SAM by sampling mixed types of prompts including bounding boxes, randomly sampled points, and coarse masks input. We generate these degraded masks by adding random Gaussian noise in the boundary regions of the GT masks. For generalizability to different object scales, we use large-scale jittering [13]. We use a learning rate of 0.001 and train our HQ-SAM for 12 epochs, with a learning rate drop after 10 epochs. We train on 8 Nvidia GeForce RTX 3090 GPUs with a total batch size of 32, which takes 4 hours to train for 16.6K iterations. Please refer to our supplemental file for more details.

**HQ-SAM Inference**  We follow the same inference pipeline of SAM but use the mask prediction from HQ-Output token as high-quality mask prediction. During inference, we sum the predicted logits of the SAM mask (by Output Token) and our predicted mask (by HQ-Output Token) for mask correction on spatial resolution $256\times256$. Then we up-sample the corrected mask to the original resolution $1024\times1024$ as our output.

**SAM vs. HQ-SAM on Training and Inference**  In Table 1, we report detailed training and inference comparisons between our HQ-SAM and SAM. While HQ-SAM produces substantially better segmentation quality, its training is very quick and affordable, which only takes 4 hours with 8 RTX3090 GPUs. HQ-SAM is also lightweight and efficient, introducing negligible increases in model parameters, GPU memory usage, and inference time per image.

Table 1: Training and inference comparison between ViT-L [11] based SAM and HQ-SAM. HQ-SAM brings negligible extra computation burden to SAM, with *less than 0.5% increase* in model parameters and reaching 96% of its original speed. SAM-L is trained on 128 A100 GPUs for 180k iterations. Based on SAM-L, we only need to train our HQ-SAM on 8 RTX3090 GPUs for 4 hours.

| Method | Training | | | | Inference | |
| | Learnable Params (M) | # GPU | Batch Size | Time (h) | FPS | Mem. |
|---|---|---|---|---|---|---|
| SAM [21] | 1191 | 128 | 128 | N/A | 5.0 | 7.6G |
| HQ-SAM | **5.1** | **8** | **32** | **4** | **4.8** | **7.6**G |

# 4 Experiments

## 4.1 Experimental Setup

**Datasets** For training we use the compiled HQSeg-44K, described in Section 3.3. For a comprehensive evaluation of the segmentation performance of HQ-SAM, we perform experiments on a wide range of datasets, including four extremely fine-grained segmentation datasets: DIS [35] (validation set), ThinObject-5K [29] (test set), COIFT [29] and HR-SOD [51]. Besides, we experiment on popular and challenging benchmarks across various image/video-based segmentation tasks in zero-shot settings, such as COCO [31], SGinW [58], UVO [42], LVIS [14], HQ-YTVIS [20] and BIG [6].

**Evaluation Metrics** To accurately quantify improvements in mask quality, instead of only employing the standard mask AP or mask mIoU, we also adopt boundary metrics mBIoU and boundary $AP_B$ [5]. We also evaluate on stricter $AP_B^{\text{strict}}$ by adjusting the default dilation ratio from 0.02 to 0.01 on UVO [42] and LVIS [14]. For evaluation on the four fine-grained segmentation datasets [35, 29, 51], we also report the averaged boundary and mask IoU among them. For video instance segmentation evaluation on HQ-YTVIS [20], we use both Tube Boundary $AP^B$ and Tube Mask $AP^M$.

## 4.2 Ablation Experiments

We conduct detailed ablation studies on the proposed HQ-SAM using ViT-Large as the backbone, analyzing the impact of the proposed HQ-Output Token and HQ-Features on segmentation quality especially in zero-shot cases. For ablation experiments, we use the four aforementioned extremely accurate segmentation datasets, namely, DIS (val) [35], ThinObject-5K (test) [29], COIFT [29] and HR-SOD [51] as well as the COCO validation set.

**Effect of the High-Quality Output Token** . HQ-SAM employs HQ-Output Token for high-quality mask prediction. Table 2 compares our HQ-Output Token to the baseline SAM and other existing prompt/token learning strategies, such as adding an additional three context tokens [56] as learnable vectors into the SAM's mask decoder for better context learning. Compared to using context tokens, the HQ-Output token consistently brings larger performance gains on four high-quality datasets, with 13.2 mBIoU on DIS and 2.7 mBIoU on COIFT datasets. We also perform other ablation experiment variants, such as computing the scaled dot product [18] between the original SAM's output token and our HQ-Output token or restricting the mask loss to only inside the boundary regions, and find they slightly decrease the averaged performance on the four evaluation datasets. Compared to SAM, HQ-SAM significantly improves the mBIoU on DIS benchmark from 52.8 to 70.4 and also promotes the mBIoU on the HRSOD dataset for 3.8 points.

**Ablation on the Global-local Fusion for HQ-Features** Table 3 tabulates the effect of global-local fusion, where the importance of each feature component is analyzed in HQ-Features during the fusion process. Compared to directly using the mask decoder feature of SAM, the entire HQ-Features bring an obvious advantage of 2.6 mBIoU on four highly accurate segmentation datasets. The final-layer ViT encoder feature with global context increases the mBIoU from 80.1 to 81.3. while the early-layer feature with local details further promotes the mBIoU to 81.8. We also replace the proposed global-local fusion with the conventional FPN to build a feature pyramid for fusion, and found this brought an inferior performance, decreasing from 89.1 to 87.4 mIoU.

**Comparison to SAM finetuning or post-refinement** . In Table 4, we compare our efficient token adaptation strategy to adding an extra post-refinement network [6] and model finetuning, including directly finetuning SAM's mask decoder or only finetuning its output token for mask prediction. Adding an extra heavy post-refinement network brings limited averaged performance increase on

Table 2: Ablation study of the HQ-Output Token on four extremely fine-grained segmentation datasets. We adopt the boxes converted from their GT masks as the box prompt input. By default, we train the predicted mask of HQ Output-Token by computing full GT mask loss.

| Model | DIS [35] mIoU | DIS [35] mBIoU | COIFT [29] mIoU | COIFT [29] mBIoU | HRSOD [51] mIoU | HRSOD [51] mBIoU | ThinObject [29] mIoU | ThinObject [29] mBIoU | Average mIoU | Average mBIoU |
|---|---|---|---|---|---|---|---|---|---|---|
| SAM (**baseline**) | 62.0 | 52.8 | 92.1 | 86.5 | 90.2 | 83.1 | 73.6 | 61.8 | 79.5 | 71.1 |
| *Using SAM's mask decoder feature*: | | | | | | | | | | |
| SAM + Context Token [56] | 71.5 | 62.2 | 93.0 | 87.7 | 91.8 | 85.0 | 84.5 | 73.1 | 85.2 | 77.0 |
| SAM + HQ-Output Token (× Output Token) | 75.1 | 65.8 | 93.9 | 88.9 | 93.0 | 86.1 | 86.1 | 74.6 | 87.0 | 78.9 |
| SAM + HQ-Output Token (Boundary Loss) | 75.2 | 66.4 | 94.0 | 88.9 | 92.1 | 85.7 | 87.3 | 76.0 | 87.2 | 79.3 |
| SAM + HQ-Output Token | 75.3 | 66.0 | 94.2 | 89.2 | 93.0 | 86.1 | 86.8 | 75.4 | 87.3 | 79.2 |
| *Using Our HQ-Feature*: | | | | | | | | | | |
| SAM + HQ-Output Token (+ Context Token) | 78.5 | 70.4 | 94.6 | 89.6 | 93.6 | **87.0** | 88.9 | 79.3 | 88.9 | 81.6 |
| SAM + HQ-Output Token | **78.6** | **70.4** | **94.8** | **90.1** | **93.6** | 86.9 | **89.5** | **79.9** | **89.1** | **81.8** |

Table 3: Ablation study on the HQ-Features sources. Early-layer denotes the feature after the first global attention block of the ViT encoder, while final-layer denotes the output of the last ViT block. Four HQ datasets denote DIS (val) [35], ThinObject-5K (test) [29], COIFT [29] and HR-SOD [51].

| Model | Fusion conv | Decoder Mask feature | ViT Encoder Final-layer | ViT Encoder Early-layer | Four HQ datasets mIoU | Four HQ datasets mBIoU |
|---|---|---|---|---|---|---|
| SAM [21] | | ✓ | | | 79.5 | 71.1 |
| HQ-SAM (Ours) | | ✓ | | | 87.3 | 79.2 |
| | ✓ | ✓ | | | 87.8 | 80.1 |
| | ✓ | | ✓ | | 15.1 | 9.0 |
| | ✓ | ✓ | ✓ | | 88.6 | 81.3 |
| | ✓ | ✓ | ✓ | ✓ | 88.6 | 81.1 |
| | ✓ | ✓ | ✓ | ✓ | **89.1** | **81.8** |

four HQ datasets but leads to very poor performance on COCO, indicating strong overfitting. We also observe a similar phenomenon when directly finetuning SAM's mask decoder. Only finetuning SAM's output token can address the catastrophic forgetting problem with improvement on the four HQ datasets and COCO. However, the incremental improvement is still much smaller compared to ours. HQ-SAM improves 1.1 $AP_B$ on COCO while output token finetuning only gives an increase of 0.4 $AP_B$. This shows the advantage of HQ-SAM in data-efficient learning while preserving the zero-shot capability of SAM.

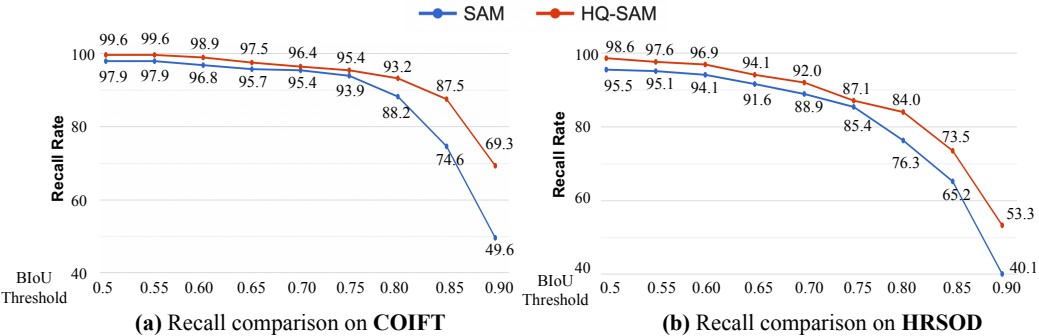

Figure 4: Recall rate comparison between COIFT [29] and HRSOD [51] under the zero-shot protocol, using BIoU thresholds ranging from loose to strict. The performance gap between SAM and our HQ-SAM increases significantly when we vary from a loose BIoU threshold of 0.5 to a very strict threshold of 0.9, showing the advantage of HQ-SAM in predicting very accurate segmentation masks.

**Accuracy analysis at different BIoU thresholds** Figure 4 compares SAM and HQ-SAM from loose to strict BIoU thresholds. We plot the percentage of mask predictions that have a BIoU larger than the threshold indicated on the x-axis. The large performance gap with strict IoU thresholds on both COIFT [29] and HRSOD [51] clearly validates the advantage of HQ-SAM in predicting very accurate masks. However, even at the loose threshold of 0.5, HQ-SAM reduces the number of incorrect predictions by SAM by 81% for COIFT and 69% for HRSOD. This shows that HQ-SAM predictions are not only substantially more accurate but also more robust in challenging cases.

Table 4: Comparison with model finetuning or extra post-refinement [6]. For the COCO dataset, we use a SOTA detector FocalNet-DINO [53] trained on the COCO dataset as our box prompt generator.

| Model | Four HQ datasets | | COCO | | | | |
|---|---|---|---|---|---|---|---|
| | mIoU | mBIoU | $AP_B$ | AP | $AP_L$ | $AP_M$ | $AP_S$ |
| SAM (baseline) | 79.5 | 71.1 | 33.3 | 48.5 | 63.9 | 53.1 | **34.1** |
| Training the whole SAM | 38.0 | 12.2 | 0.2 | 5.5 | - | - | - |
| Add Context Token [56] | 85.2 | 77.0 | 31.9 | 47.2 | 65.1 | 51.2 | 31.9 |
| CascadePSP Post-refinement [6] | 80.9 | 74.6 | 2.8 | 13.4 | 43.4 | 9.4 | 0.0 |
| CRM Post-refinement [37] | 81.4 | 75.4 | 15.9 | 28.7 | - | - | - |
| Finetune SAM's decoder | 87.6 | 79.5 | 9.0 | 19.5 | 45.2 | 15.8 | 4.7 |
| Finetune SAM's output token | 87.6 | 79.7 | 33.7 | 48.7 | 66.0 | 52.3 | 33.6 |
| HQ-SAM (Ours) | **89.1** | **81.8** | **34.4** | **49.5** | **66.2** | **53.8** | 33.9 |

Table 5: Zero-shot open-world instance segmentation results comparison on UVO [42]. We use FocalNet-DINO [53] trained on the COCO dataset as our box prompt generator. $*^{strict}$ denotes the boundary region with a tighter threshold.

| Model | $AP_B^{strict}$ | $AP_{B75}^{strict}$ | $AP_{B50}^{strict}$ | $AP_B$ | $AP_{B75}$ | $AP_{B50}$ | AP |
|---|---|---|---|---|---|---|---|
| SAM | 8.6 | 3.7 | 25.6 | 17.3 | 14.4 | 37.7 | 29.7 |
| HQ-SAM | **9.9** | **5.0** | **28.2** | **18.5** | **16.3** | **38.6** | **30.1** |

Table 6: Zero-shot segmentation result comparison on the test set of high-quality BIG [6] benchmark using various types of input prompts. We employ PSPNet [55] to generate the coarse mask prompt.

| Model | GT Box Prompt | | Mask Prompt | |
|---|---|---|---|---|
| | mIoU | mBIoU | mIoU | mBIoU |
| SAM | 81.1 | 70.4 | 66.6 | 41.8 |
| HQ-SAM | **86.0** | **75.3** | **86.9** | **75.1** |

## 4.3 Zero-shot Comparison with SAM

We perform extensive zero-shot transfer comparisons between our HQ-SAM and SAM on 7 benchmarks, including SGinW [58], COCO [31], UVO [42], LVIS [14], HQ-YTVIS [20], BIG [6], COIFT [29] and HR-SOD [51], where HQ-SAM outperforms SAM without bells and whistles, demonstrating its efficacy and kept generalization ability even trained with a small-scale dataset.

**Results on the SGinW Benchmark** Equipped with the same Grounding-DINO [32] as box prompts, we also performed experiments by replacing SAM with HQ-SAM in Grounded-SAM, and obtained **the first place** in the Segmentation in the Wild (SGinW) competition[1] on the zero-shot track. Note that SGinW contains *25 zero-shot in-the-wild segmentation datasets* for evaluation, and Grounded-HQ-SAM with 49.6 mean AP and outperforms Grounded-SAM obviously using the same detector.

**Zero-Shot Open-world Segmentation** To evaluate the zero-shot segmentation results in the open-world environment, in Table 5, we compare SAM and our HQ-SAM on the challenging UVO [42] benchmark with diverse and dense objects mask annotations. By taking the same pre-trained object detector [53] as box prompt input, our HQ-SAM improves for 1.3 $AP_B^{strict}$ and 2.6 $AP_{B50}^{strict}$ over SAM.

**Zero-Shot Segmentation on High-resolution BIG Dataset** In Table 6, we compare the zero-shot segmentation quality between SAM and HQ-SAM on the high-resolution BIG benchmark [6] with two types of prompts, including using GT object boxes or the provided coarse masks input. HQ-SAM consistently surpasses SAM, with obvious advantages using different types of prompts, and is much more robust to coarse masks prompts with partial boundary errors (provided by PSPNet [55]).

**Zero-shot Instance Segmentation on COCO and LVIS** In Table 7, we also evaluate HQ-SAM on the popular COCO and LVIS benchmarks respectively by feeding box prompts generated by the trained detectors of these two datasets. HQ-SAM consistently outperforms SAM by 1.1 $AP_B$ on COCO and 0.7 $AP_{B75}^{strict}$ on LVIS, showing the improved mask quality and well-preserved zero-shot segmentation ability during the HQ-SAM training process.

---

[1] SGinW Benchmark Results: https://eval.ai/web/challenges/challenge-page/1931/leaderboard/4567

Table 7: Zero-shot instance segmentation results comparison on COCO [31] and LVISv1 [14]. For the COCO dataset, we use FocalNet-DINO [53] detector trained on COCO. For LVIS, we adopt ViTDet-H [28] trained on the LVIS dataset as our box prompt generator. For SAM, we use the ViT-L backbone and box prompt. We maintain the zero-shot segmentation capability of the original SAM while improving the mask quality on the boundary region.

| Model | COCO | | LVIS | | | | |
| | $AP_B$ | AP | $AP_B^{strict}$ | $AP_{B75}^{strict}$ | $AP_B$ | $AP_{B75}$ | AP |
|---|---|---|---|---|---|---|---|
| SAM | 33.3 | 48.5 | 32.1 | 32.8 | 38.5 | 40.9 | 43.6 |
| HQ-SAM | **34.4** | **49.5** | **32.5** | **33.5** | **38.8** | **41.2** | **43.9** |

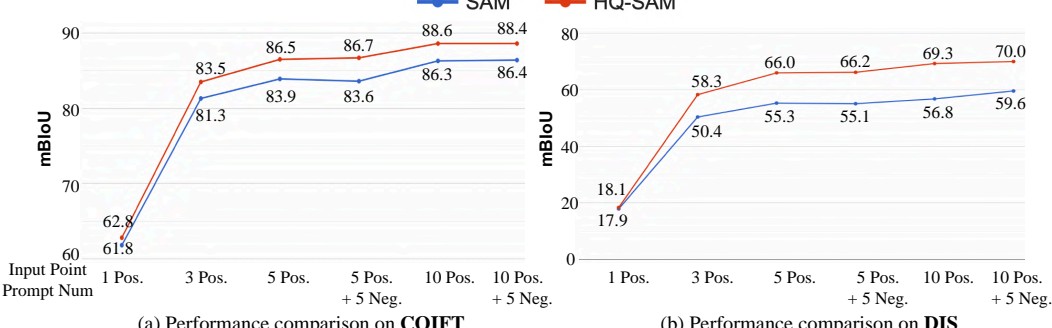

(a) Performance comparison on **COIFT**      (b) Performance comparison on **DIS**

Figure 5: Interactive segmentation results comparison using a varying number of input points on the COIFT [29] (zero-shot) and DIS [35] val set. HQ-SAM consistently outperforms SAM with various point numbers, and the relative improvement is more obvious with less prompt ambiguity.

Table 8: Zero-shot Video Instance Segmentation comparison on the test set of the very accurately labeled HQ-YTVIS [20] benchmark. We utilize pre-trained Swin-L-based Mask2Fromer [4] on YTVIS [47] as our box prompt input while reusing its object association prediction.

| Model | $AP^B$ | $AP_{75}^B$ | $AP_{50}^B$ | $AP^M$ | $AP_{75}^M$ | $AP_{50}^M$ |
|---|---|---|---|---|---|---|
| SAM | 30.2 | 19.1 | 72.9 | 60.7 | 68.1 | 90.5 |
| HQ-SAM | **34.0** | **24.3** | **79.5** | **63.6** | **70.5** | **91.1** |

**Point-based Interactive Segmentation Comparison** To investigate the segmentation performance of HQ-SAM with interactive point prompts, in Figure 5, we compare HQ-SAM to SAM with varying numbers of input points on COIFT [29] (zero-shot) and DIS [35] val set. HQ-SAM consistently outperforms SAM with different point prompts on both two datasets. We note that the relative performance increase is more significant when the prompt contains less object ambiguity with more input points information (increasing from 1 positive point to 10 positive points + 5 negative points).

**Zero-shot High-quality Video Instance Segmentation** Besides conducting image-based segmentation evaluation, we also perform video instance segmentation results comparison on the accurately annotated HQ-YTVIS benchmark [20]. We take the pre-trained Mask2Former [4] as our video box prompts and feed it into SAM and our HQ-SAM for mask prediction. In Table 8, HQ-SAM achieves remarkable gains of 3.8 points in Tube Boundary $AP^B$ and 2.9 Tube Mask $AP^M$.

**Visualization of HQ-Output Token** In Figure 6, we provide visual comparison of our HQ-Output Token vs. SAM's common output token for their cross-attention maps in the last token-to-image layer of the mask decoder. We observe that our HQ-Output Token attends to the boundary and thin structure regions that are missed by the common token.

**Zero-shot Visual Results Comparison** In Figure 7, we compare HQ-SAM to SAM qualitatively in a zero-shot transfer setting, where HQ-SAM significantly promotes the mask details of SAM and also improves the masks of broken holes or large portion errors by the enriched semantic context. Refer to the supplemental file for more visual comparisons.

**Comparison with Adapter Tuning Strategy** In Table 9, we also compare our efficient token adaptation strategy to the recent Adapter Tuning [48] and LoRA [17]. We introduce lightweight adapters to ViT layers of SAM's encoder for encoder tuning and identify that this strategy leads to overfitting and its zero-shot performance on COCO decreases from 33.3 to 29.6. This validates our design choice to freeze SAM's encoder, and mainly focus on SAM's decoder.

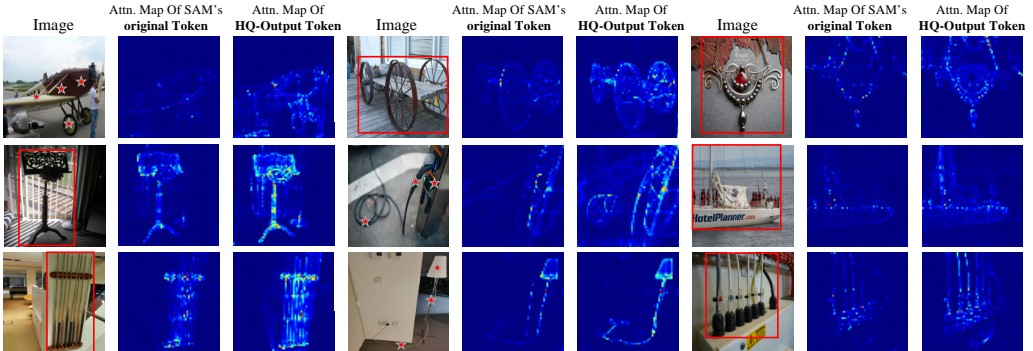

Figure 6: Cross-attention of SAM's original token **vs.** HQ-Output Token in the last decoder layer. HQ-Token attends to the boundary and thin structure regions that are missed by the original token.

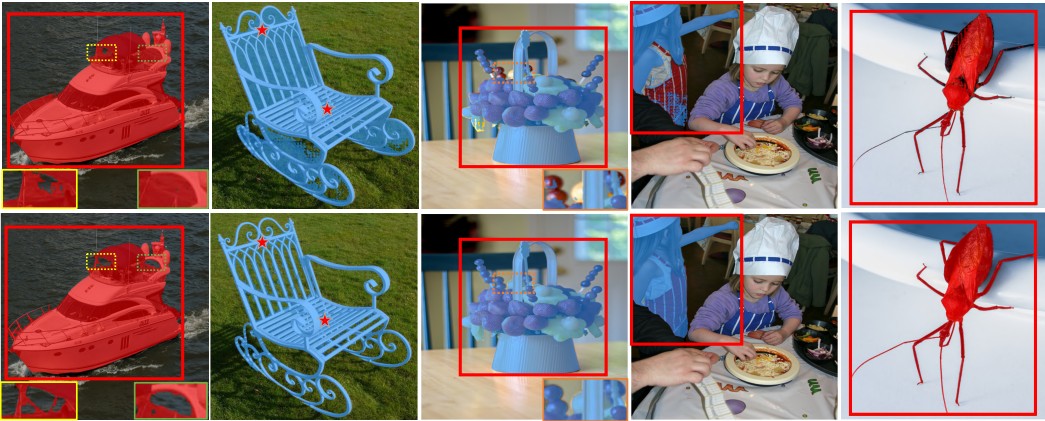

Figure 7: Visual results comparison between SAM (top row) **vs.** HQ-SAM (bottom row) in a *zero-shot transfer setting*, given the same red box or point prompt. HQ-SAM produces significantly more detailed-preserving results and also addresses the mask errors with broken holes.

Table 9: Comparison to Adapter Tuning [48] or using LoRA [17] in SAM's encoder using ViT-L based SAM and the same HQSeg-44K. For the COCO dataset, we use the SOTA detector FocalNet-DINO [53] trained on the COCO dataset as our box prompt generator.

| Model | COCO | | | | | Model Params (MB) | |
| | $AP_B$ | AP | $AP_L$ | $AP_M$ | $AP_S$ | Total | Trainable |
|---|---|---|---|---|---|---|---|
| SAM | 33.3 | 48.5 | 63.9 | 53.1 | 34.1 | 1191 | - |
| SAM + LoRA [17] | 28.6 | 43.7 | - | - | - | 1192.5 | 1.5 |
| SAM + Encoder Adapter [48] | 29.6 | 44.8 | 63.9 | 47.8 | 29.0 | 1203 | 12.0 |
| HQ-SAM | **34.4** | **49.5** | 66.2 | 53.8 | 33.9 | 1196.1 | 5.1 |

**Mobile Efficiency** Although HQ-SAM significantly boosts SAM's mask quality with negligible overhead, it shares the heavy ViT encoder of SAM, and thus cannot achieve a real-time speed in video processing. For efficient mobile deployment, we propose Light HQ-SAM based on the tiny ViT image encoder provided by MobileSAM [52]. In Figure 2, achieving running speed of 41.2 FPS, Light HQ-SAM improves the zero-shot COCO AP of MobileSAM from 44.3 to 45.0 with negligible additional cost, i.e., 1.7MB increase in model parameters.

## 5 Conclusion

We propose HQ-SAM, the first high-quality zero-shot segmentation model by introducing negligible overhead to the original SAM. We propose a lightweight High-quality Output Token in HQ-SAM to replace the original SAM's output token for high-quality mask prediction. After training only on 44K highly-accurate masks, HQ-SAM significantly boosts the mask prediction quality of SAM, which was trained on 1.1 billion masks. The zero-shot transfer evaluation is performed on 8 segmentation benchmarks across both image and video tasks, spanning diverse objects and scenes. Our research offers timely insights into how to leverage and extend SAM-like foundational segmentation models in a data-efficient and computation-affordable manner.

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

# Supplementary Material:
# Segment Anything in High Quality

In this supplementary material, Section 6 first presents the additional experimental analysis of our HQ-SAM, including more zero-shot transfer comparisons to SAM on both image and video benchmarks. Then, in Section 7, we describe more details of our method implementation, including the training and inference. In Section 8, we provide further details of our constructed HQSeg-44K dataset for training HQ-SAM. In Section 9, we show extensive visual results comparison between our HQ-SAM and SAM on COCO [31], DIS-test [35], HR-SOD [51], NDD20 [41], DAVIS [34], and YTVIS [47].

## 6 Supplementary experiments

**SAM vs. HQ-SAM on Various Backbones** In Table 10, we provide a comprehensive comparison between HQ-SAM and SAM using various backbones, including ViT-B, ViT-L, ViT-H and TinyViT. The comparison not only includes the numerical results on the four HQ datasets and COCO validation set, but also contains the model sizes/speed/memory. HQ-SAM consistently outperforms SAM using three different backbones, with over 10 points increase in mBIoU on the four HQ datasets. Notably, the ViT-B based HQ-SAM significantly improves the $AP^B$ on COCO from 28.2 to 31.3 and AP from 44.4 to 46.7, with only a 1.1% increase in model parameters and negligible extra memory consumption.

Table 10: SAM vs. HQ-SAM on various ViT backbones. For the COCO dataset, we use a SOTA detector FocalNet-DINO [53] trained on the COCO dataset as our box prompt generator.

| Model | Four HQ datasets | | COCO | | | | | Model Params (MB) | | FPS | Memory |
|---|---|---|---|---|---|---|---|---|---|---|---|
| | mIoU | mBIoU | $AP_B$ | AP | $AP_L$ | $AP_M$ | $AP_S$ | Total | Learnable | | |
| SAM-B | 70.6 | 62.3 | 28.2 | 44.4 | 57.7 | 48.7 | 32.1 | 358 | 358 | 10.1 | 5.1G |
| HQ-SAM-B | **86.3** | **78.1** | **31.3** | **46.7** | 62.9 | 50.5 | 32.0 | 362.1 | **4.1** | 9.8 | 5.1G |
| SAM-L | 79.5 | 71.1 | 33.3 | 48.5 | 63.9 | 53.1 | 34.1 | 1191 | 1191 | 5.0 | 7.6G |
| HQ-SAM-L | **89.1** | **81.8** | **34.4** | **49.5** | 66.2 | 53.8 | 33.9 | 1196.1 | **5.1** | 4.8 | 7.6G |
| SAM-H | 75.6 | 68.3 | 34.0 | 48.9 | 64.5 | 53.3 | 34.4 | 2446 | 2446 | 3.5 | 10.3G |
| HQ-SAM-H | **89.3** | **81.5** | **34.9** | **49.9** | 66.5 | 54.0 | 34.2 | 2452.1 | **6.1** | 3.4 | 10.3G |
| MobileSAM | 69.0 | 58.8 | 28.6 | 44.3 | - | - | - | 38.6 | 38.6 | 44.8 | 3.7G |
| Light HQ-SAM | **81.4** | **71.6** | **29.6** | **45.0** | - | - | - | 40.3 | **1.7** | 41.2 | 3.7G |

Table 11: Results on YouTubeVIS 2019 validation set and HQ-YTVIS test set using ViT-L based SAM. We adopt the SOTA detector Mask2Former [4] trained on the YouTubeVIS 2019 dataset as our video boxes prompt generator while reusing its object association prediction.

| Model | YTVIS 2019 | | | | | | HQ-YTVIS | |
|---|---|---|---|---|---|---|---|---|
| | AP | $AP_{50}$ | $AP_{75}$ | $AP_L$ | $AP_M$ | $AP_S$ | $AP^B$ | $AP^M$ |
| SAM | 51.8 | 82.1 | 55.4 | 65.5 | 52.0 | 34.2 | 30.2 | 60.7 |
| HQ-SAM | **53.2** | 82.9 | 58.3 | 66.4 | 53.3 | 33.7 | **34.0** | **63.6** |

**Zero-shot Video Instance Segmentation Comparison** Extending from Table 8 of the paper (evaluation on the HQ-YTVIS benchmark [20]), we further perform a comparative analysis of zero-shot video instance segmentation results on the popular YTVIS 2019 [47] validation set. We take the pre-trained Mask2Former [4] as our video box prompts and feed them into SAM and our HQ-SAM for mask prediction. In Table 11, HQ-SAM achieves consistent gains of 1.4 points in Tube Mask AP, increasing SAM's performance from 51.8 to 53.2. Interestingly, we find the $AP_{75}$ improvement with a higher IoU threshold for HQ-SAM is much larger than $AP_{50}$, further validating the advantages of HQ-SAM in high-quality mask prediction.

**Zero-shot Video Object Segmentation Comparison**  Besides video instance segmentation, in Table 12, we further report the comparison of video object segmentation results between HQ-SAM and SAM on DAVIS validation set in a zero-shot transfer protocol. We take the pre-trained XMem as our video box prompts and feed the same prompts into SAM and HQ-SAM. HQ-SAM improves SAM the $\mathcal{J}\&\mathcal{F}$ from 82.0 to 83.2 and the $\mathcal{F}$ score from 84.9 to 86.1, where $\mathcal{F}$ is for measuring the contour accuracy of the video objects.

Table 12: Results on DAVIS 2017 [34] validation set using ViT-L based SAM. We adopt the SOTA model XMem [7] as our video boxes prompt generator while reusing its object association prediction.

| Model | $\mathcal{J}\&\mathcal{F}$ | $\mathcal{J}$ | $\mathcal{F}$ |
|---|---|---|---|
| SAM | 82.0 | 79.0 | 84.9 |
| HQ-SAM | **83.2** | **80.3** | **86.1** |

**Robustness to Input Box Prompts**  In Table 13, we compare HQ-SAM to SAM by adding various scales of noises to the input ground truth box prompts. In practice, we cannot expect the input box prompts provided by humans in interactive modes to be identical to the ground truth (GT) boxes or extremely accurate. We follow the data augmentation code in DN-DETR [25] to add different noise scales and identify that our HQ-SAM is much more robust compared to SAM, where the relative mBIoU advantage improves from 10.7 to 20.5 when gradually increasing the noise scales. Note that our method is not trained with noised boxes. We also visualize such noised input case in Figure 11, where SAM is more sensitive to small box location shifts that easily happened during interactive annotation.

Table 13: Comparison of segmentation accuracy on the four HQ datasets by adding various noise levels to the GT box prompts input.

| Model | No Noise | | Noise scale 0.2 | | Noise scale 0.4 | |
|---|---|---|---|---|---|---|
| | mIoU | mBIoU | mIoU | mBIoU | mIoU | mBIoU |
| SAM | 79.5 | 71.1 | 65.7 | 57.1 | 46.4 | 39.8 |
| HQ-SAM | 89.1 | **81.8**$_{\uparrow 10.7}$ | 82.8 | **73.4**$_{\uparrow 16.3}$ | 69.9 | **60.3**$_{\uparrow 20.5}$ |

## 7   Additional Implementation details

**Training Details**  During training HQ-SAM on the composed HQSeg-44K, we fix the model parameters of the pre-trained SAM model while only making the proposed HQ-SAM learnable, including HQ-Output Token, its associated three-layer MLP and three convolutions for HQ-Features fusion. Two of them are transposed convolutions (size $2\times2$, stride 2) used to upscale encoder embedding size from $64\times64$ to $256\times256$. We treat the new HQ-Output Token as the fifth mask token compared to the original four mask tokens in SAM's mask decoder. During training, this new HQ-Output token of size $1\times256$ is concatenated with SAM's mask tokens (size of $4\times256$), iou token (size of $1\times256$) and prompt tokens (size of $N_{prompt}\times256$) as the input to the SAM's mask decoder. For example, if the input image contains $N$ box prompts (size $N\times2\times256$), the final concatenated input and output shape for the 2-layer mask decoder of SAM is $N\times(1+4+1+2)\times256$. For experiments using ViT-B, ViT-L, and ViT-H-based models on training, we adopt the same training setting, with a learning rate of 1e-3 and train our HQ-SAM for 12 epochs (learning rate drops to 1e-4 after 10 epochs). We supervise mask prediction of the new HQ-Output token with a combination of both BCE Loss and Dice Loss.

**Implementation Details**  We follow the same inference pipeline of SAM but use the mask prediction from HQ-Output token as high-quality mask prediction. Table 10 reports the detailed inference speed comparison using various backbones. For box-prompting-based evaluation, we feed SAM and our HQ-SAM with the same image/video bounding boxes and adopt the single mask output mode of SAM. For interactive segmentation comparison using a single point, we follow SAM and adopt the "center" point of Ground Truth (GT) masks, which is at a maximal value location in a mask's interior distance transform. For multiple-point evaluation, we randomly sample the points from the GT masks and report the averaged results with three trials.

# 8 More Details of HQSeg-44K

**Data compostion of HQSeg-44K** In Table 14, we provide more details of our composed new training dataset HQSeg-44K which contains 44,320 extremely accurate image mask annotations, where we show their annotation quality in Figure 8. HQSeg-44K is a collection of six existing image datasets including DIS [35] (train set), ThinObject-5K [29] (train set), FSS [26], ECSSD [38], MSRA-10K [8], DUT-OMRON [46] with extremely fine-grained mask labeling, where each of them contains 7.4K mask labels on average. This composed training set has no images/annotations overlapping with the zero-shot evaluation datasets adopted in our paper.

**Effect of HQSeg-44K** In Table 15, we show the advantage of using HQSeg-44K by comparing HQ-SAM training with 44K randomly sampled images and masks from SA-1B [21]. Using the same efficient token learning strategy, training with SA-1B (44K) decreases the averaged mBIoU on the four datasets from 71.1 to 70.1, while ours improves it from 71.1 to 81.8. This validates the effectiveness of our constructed HQSeg-44K benchmark in improving mask quality. Note that the ablation experiments in Table 2, Table 3, Table 4, and Table 9 of the paper are all based on the constructed HQSeg-44K.

Table 14: Data composition of our constructed HQ-Seg-44K.

| Dataset | DIS [35] | Thin-Object 5k [29] | FSS [26] | DUTS [46] | ECSSD [38] | MSRA-10K [8] | Total |
|---|---|---|---|---|---|---|---|
| Image Num. | 3000 | 4748 | 10000 | 15572 | 1000 | 10000 | 44320 |

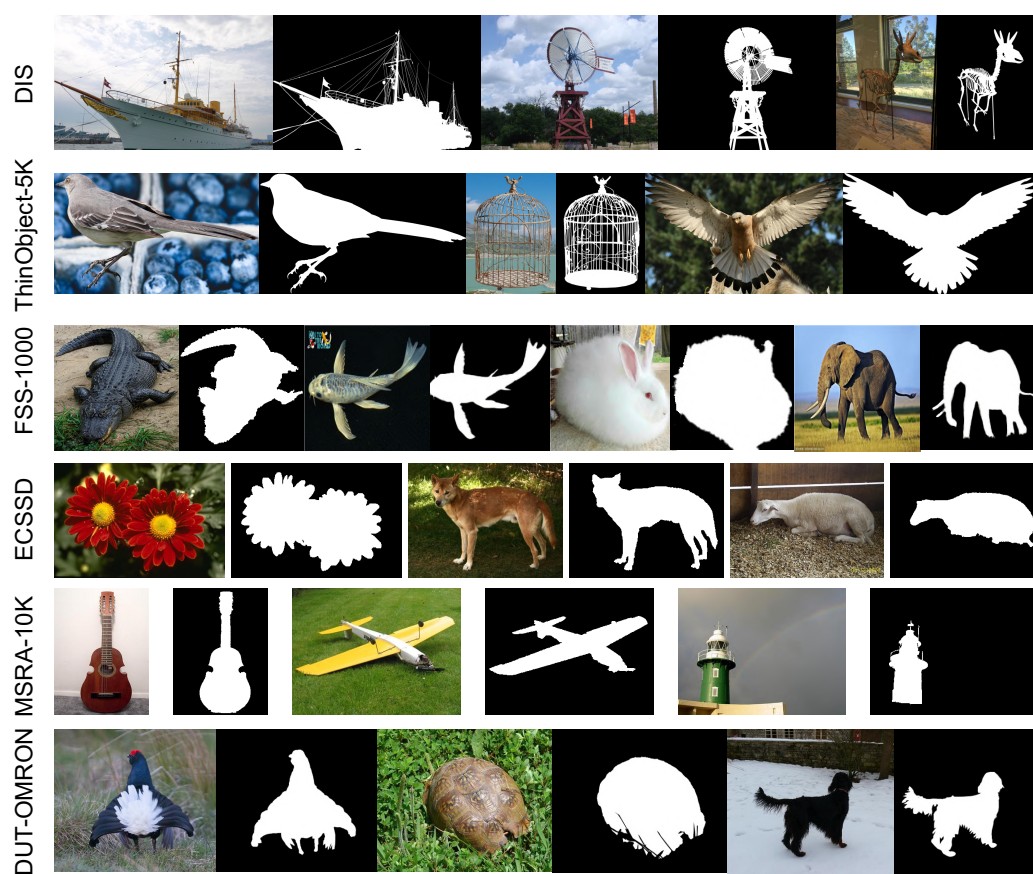

Figure 8: Visualization of annotated mask quality for randomly selected cases from the six dataset components of the HQ-Seg-44K. Zoom in for better viewing the fine-grained mask details.

**Zero-shot results on DIS and ThinObject-5K** We also report zero-shot results in Table 16 on DIS and ThinObject-5K by removing the training splits of either or both datasets from the training of

Table 15: Comparison of the training dataset. For the COCO dataset using ViT-L-based SAM, we use a SOTA detector FocalNet-DINO [53] trained on the COCO dataset as our box prompt generator.

| Model | Dataset | DIS | | COIFT | | HRSOD | | ThinObject | | Average | |
|---|---|---|---|---|---|---|---|---|---|---|---|
| | | mIoU | mBIoU | mIoU | mBIoU | mIoU | mBIoU | mIoU | mBIoU | mIoU | mBIoU |
| SAM | SA-1B | 62.0 | 52.8 | 92.1 | 86.5 | 90.2 | 83.1 | 73.6 | 61.8 | 79.5 | 71.1 |
| HQ-SAM | + SA-1B-44K | 60.4 | 51.7 | 91.1 | 86.1 | 88.4 | 80.9 | 73.1 | 61.8 | 78.3 | 70.1 |
| HQ-SAM | + HQ-Seg-44K (Ours) | 78.6 | 70.4 | 94.8 | 90.1 | 93.6 | 86.9 | 89.5 | 79.9 | 89.1 | 81.8 |

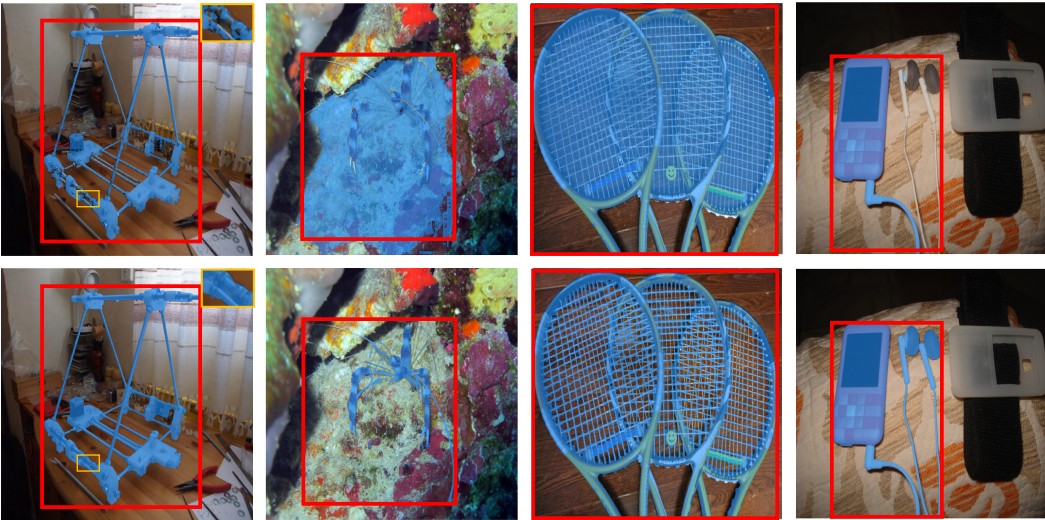

Figure 9: Visual results comparison between SAM (top row) **vs.** HQ-SAM (bottom row) on DIS test set, given the same red box prompt. HQ-SAM produces significantly more accurate boundaries.

HQ-SAM. The improvement of HQ-SAM over SAM is still substantial on DIS or ThinObject (over 10.0 points on DIS-mIoU and 9.0 points on ThinObject-mIoU), even when the corresponding training splits are removed from training.

Table 16: Zero-shot results on DIS and ThinObject-5K by removing the training splits of either or both datasets from the training of HQ-SAM. Results not obtained in a zero-shot manner (i.e. the training split was used), are shown in parenthesis to easily compare zero-shot results.

| Training Setting | DIS-mIoU | DIS-mBIoU | ThinObject-mIoU | ThinObject-mBIoU |
|---|---|---|---|---|
| SAM (**baseline**) | 62.0 | 52.8 | 73.6 | 61.8 |
| HQ-SAM (**remove both DIS and ThinObject**) | 72.9 | 63.1 | 82.7 | 70.7 |
| HQ-SAM (**remove DIS**) | 74.7 | 66.2 | (90.1) | (80.4) |
| HQ-SAM (**remove ThinObject**) | (78.4) | (70.3) | 83.3 | 72.1 |
| HQ-SAM (**default HQSeg-44K**) | (78.6) | (70.4) | (89.5) | (79.9) |

## 9 More Visual Results Comparison

We provide more extensive visual results comparison in Figure 9 (DIS [35] test set), Figure 10 (zero-shot setting in COCO), Figure 11 (noised box input) and Figure 12 (zero-shot setting in HRSOD [51], NDD20 [41] and web images which cover objects with various structure complexities in diverse environments. In Figure 13 and Figure 14, we provide the zero-shot video segmentation results comparison on DAVIS 2017 and YTVIS 2019 benchmarks respectively. Besides, we include the dark underwater environment in NDD20 [41] and randomly selected web images in Figure 12, showing that the zero-shot segmentation power in SAM is well preserved by HQ-SAM. In Figure 12, we also include two failure cases in the rightmost two columns of the third row and bottom row, where HQ-SAM improves over SAM, but still cannot achieve fully correct mask prediction.

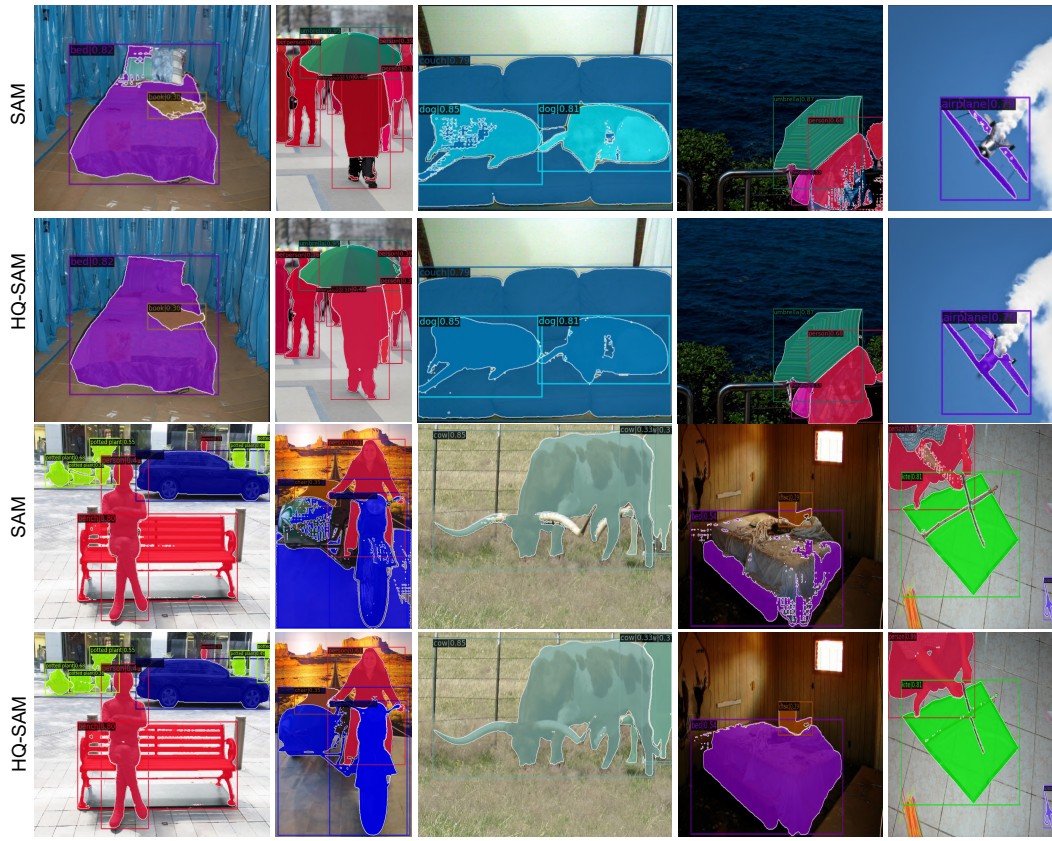

Figure 10: Visual results comparison between SAM (top row) **vs.** HQ-SAM (bottom row) on COCO val set in *zero-shot setting*, using a SOTA detector FocalNet-DINO [53] trained on the COCO dataset as our box prompt generator. HQ-SAM predicts masks with higher quality than SAM with less mask artifacts.

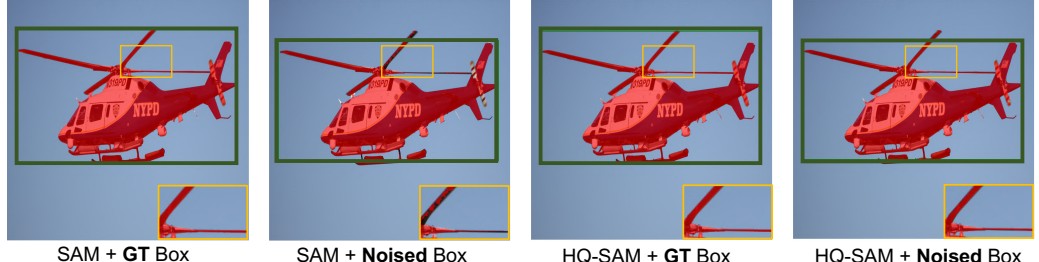

| SAM + **GT** Box | SAM + **Noised** Box | HQ-SAM + **GT** Box | HQ-SAM + **Noised** Box |

Figure 11: Visual results comparison between SAM (top row) **vs.** HQ-SAM (bottom row) with both the GT and noised green box prompt. HQ-SAM produces much more consistent and robust segmentation results regarding to the noises in the input boxes.

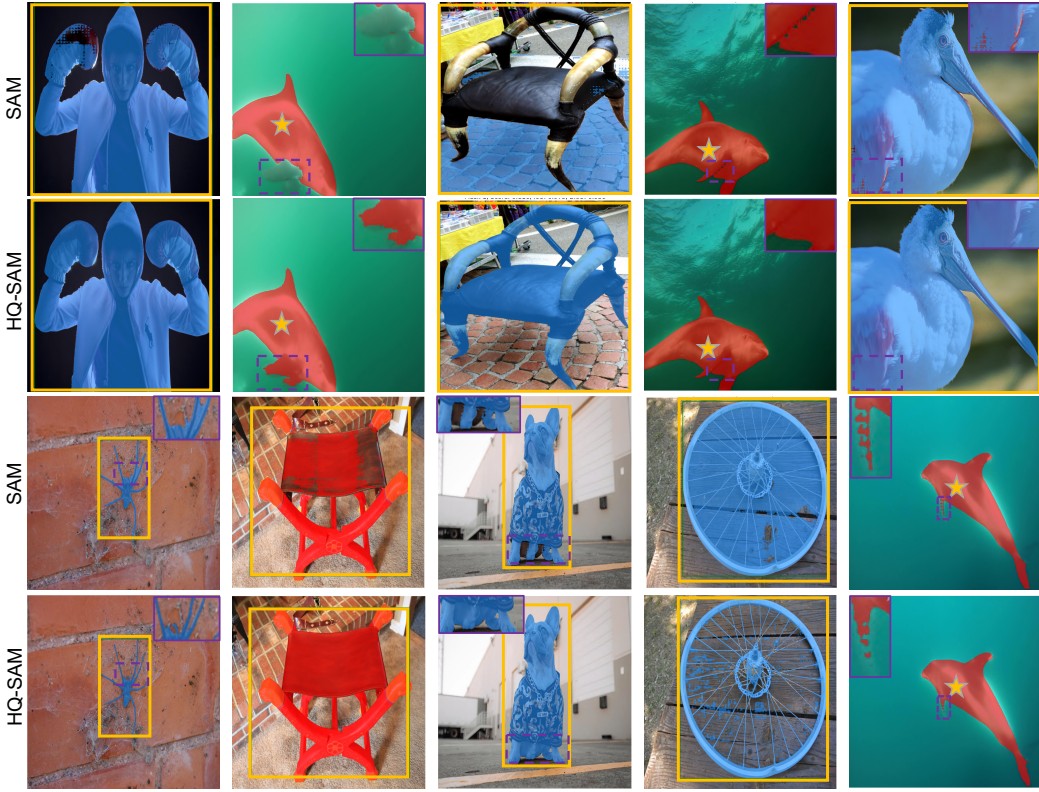

Figure 12: Visual results comparison between SAM (top row and third row) **vs.** HQ-SAM (second row and bottom row) in *zero-shot setting*, given the same yellow box or point prompt. HQ-SAM produces significantly more detailed preserving masks while fixing mask errors with broken holes. The rightmost two columns in the third row and bottom row show two *failure cases* of HQ-SAM in extremely dark environments or very tiny metal rods.

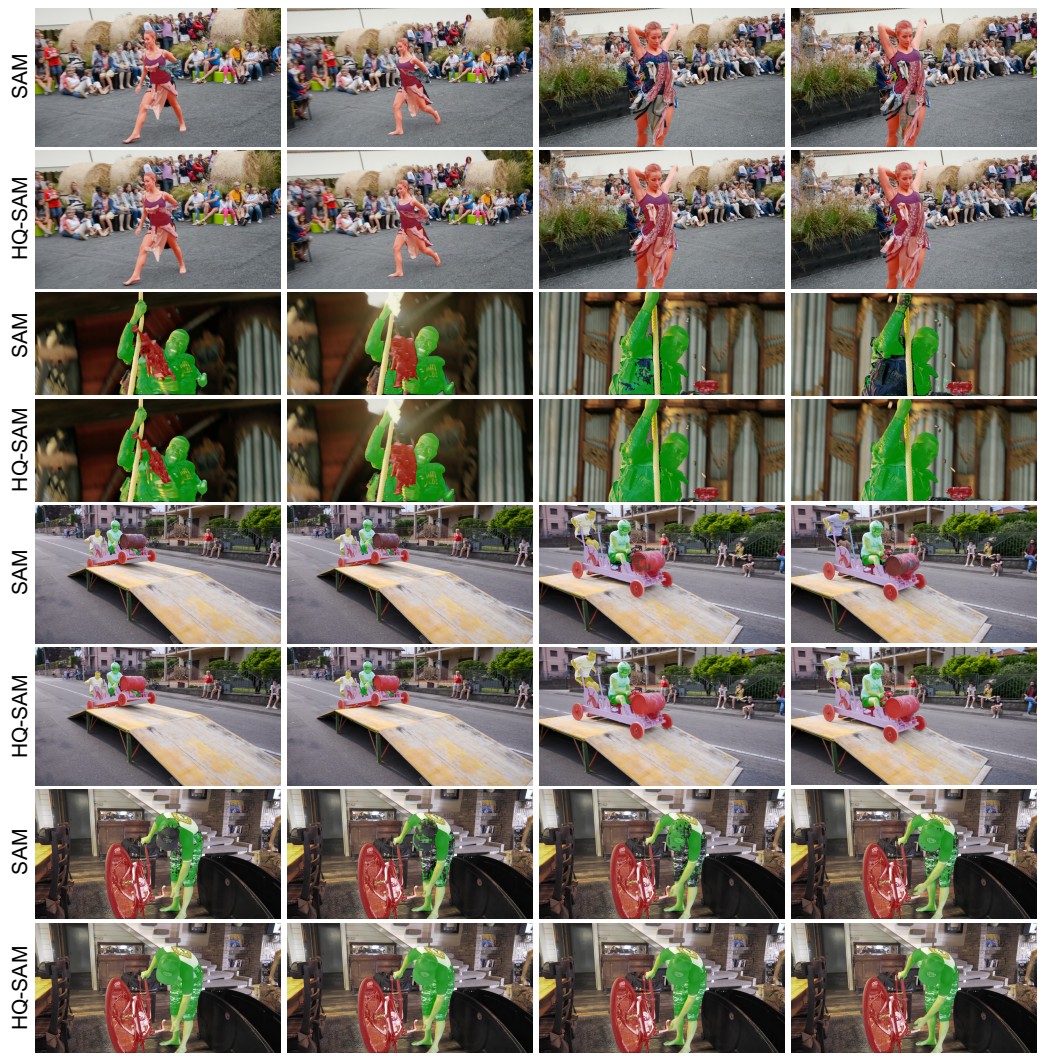

Figure 13: Visual results comparison between SAM **vs.** HQ-SAM on video object segmentation benchmark DAVIS 2017 in *zero-shot setting*, given the same video boxes prompts generated by the pre-trained XMem [7].

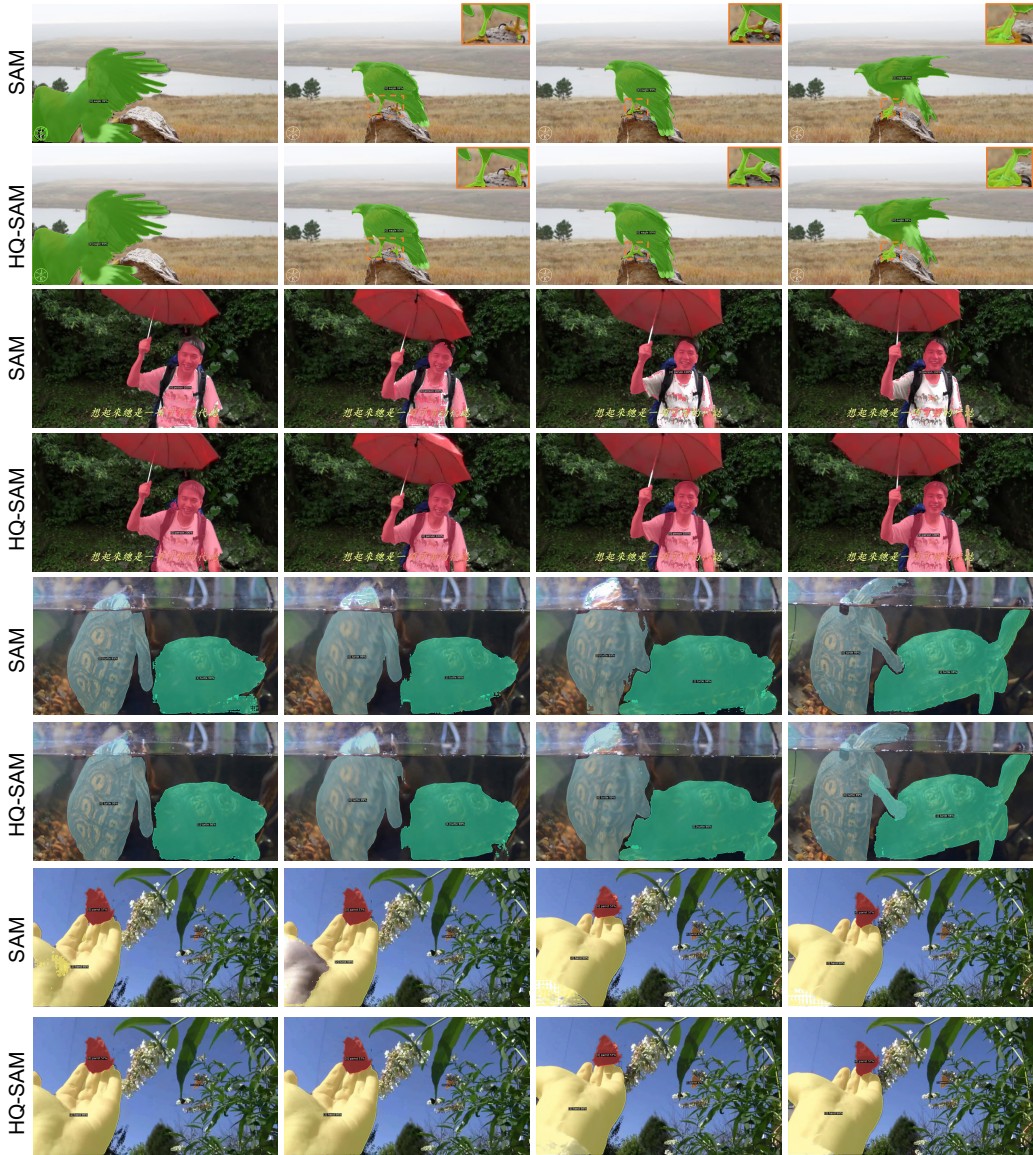

Figure 14: Visual results comparison between SAM **vs.** HQ-SAM on video instance segmentation benchmark YTVIS 2019 in *zero-shot setting*, given the same video boxes prompts generated by the pre-trained Mask2Former [4].

