# OpenReview forum: "Segment Anything in High Quality"
_NeurIPS.cc/2023/Conference — NeurIPS 2023 poster_

### Official Review · Reviewer_eEa5 · 2023-07-06

**Soundness:** 2 fair
**Presentation:** 3 good
**Contribution:** 2 fair
**Rating:** 5
**Confidence:** 4

**Summary:**

The authors observe that the segmentation results of SAM model are unsatisfactory in several cases such as coarse mask boundaries and increect predictions. Thus, this paper presents HQ-SAM, whose objective is to improve the segmentation capability of SAM, with the introduction of the learnable high-quality output token and the global-local fusion module. For the training process, the SAM model is freezed while the newly added components are trained on the combination of 6 datasets, i.e., DIS (train set), ThinObject-5K (train set), FSS-1000, ECSSD, MSRA177 10K and DUT-OMRON. Experiments on four datasets (DIS, COIFT, HRSOD and ThinObject) show the effectiveness of the proposed method. Nevertheless, using DIS and ThinObject as testing sets may be unfair when comparing with SAM that may not see these two datsets during training.

**Strengths:**

1. The motivation of improving SAM model is good. SAM is a powerful foundation model in computer vision, but its performance is not always satisfactory. This paper takes a very first step towards improving SAM.
2. Good presentation and easy to read.

**Weaknesses:**

1. From Fig.2, it seems that there is an error correction module, but I didn't see any descriptions about this module in Section 3.2.
2. The training set is composed of DIS (train set), ThinObject-5K (train set), FSS-1000, ECSSD, MSRA177 10K and DUT-OMRON, while the valuations is conducted on DIS and ThinObject, which is unfair when comparing with SAM model. From Table 2, the improvement on DIS and ThinObject is much higher that on COIFT and HRSOD. This is because the corresponding training images are used in the training. Thus,  evaluatoin on such datasets cannot verity the zero-shot generation capability of the proposed approach.
3. I encourage the authors to conduct experiments without using error correction during inference on COCO, LVIS, COIFT and HRSOD .

**Questions:**

1. Why do authors select these 6 datasets to form HQSeg-44K benchmark? What's the motivation behind this kind of selection? Please also see weaknesses box for other questions. I hope the authors could address my concerns in rebuttal.

**Limitations:**

As authors claimed, HQ-SAM shares the heavy ViT encoder of SAM, and thus cannot achieve a real-time speed in video processing.

---

> ### Author Rebuttal · Authors · 2023-08-08
>
> We thank the reviewer for acknowledging our paper being the first step to improve SAM, with good motivation and presentation. Here we respond to your insightful questions.
>
> > **Q1.** What’s the error correction module?
>
> Error correction is not a module, but simply a direct element-wise sum between the predicted logits of the SAM (by Output Token) and predicted mask by HQ-Output Token. While this is stated in L193-L195 of paper, we agree that it is not clearly connected to the term "Error correction" used in Fig. 2. We will clarify this in next revision by updating Fig.2 and its caption. The element-wise sum operation has no model parameters. Obtaining high-quality mask using coarse mask as input is common in high-quality segmentation works [21, 6, 18] for mask correction.
>
> > **Q2.** The zero-shot evaluation experiments.
>
> As stated in the abstract and L63-L64 of paper, we performed zero-shot comparison with SAM on **7 zero-shot benchmarks** including COCO, UVO, LVIS, HQ-YTVIS, BIG, COIFT and HR-SOD. In the Supp. file, we further provided zero-shot comparison with SAM on popular Youtube-VIS (Tab. 3 of the Supp.) and DAVIS (Tab. 4 of the Supp.). HQ-SAM consistently outperforms SAM in all these zero-shot evaluation experiments.
>
> In addition, we evaluated HQ-SAM in the **SGinW** (Smentation in the Wild) competition, and obtained the **first place** on zero-shot track (hosted in CVPR 2023). Note SGinW benchmark adopts **25** zero-shot in-the-wild segmentation datasets as evaluation. Using the same Grounding-DINO detector as box prompt, Grounded-HQ-SAM outperforms Grounded-SAM obviously. The SGinW benchmark leaderboard is public.
>
> Besides, we also report zero-shot results here on DIS and ThinObject-5K by removing the training splits of either or both datasets from the training of HQ-SAM. Results not obtained in a zero-shot manner (i.e. the training split was used), are shown in parenthesis to easily compare zero-shot results.
>
> |  Training Setting  |  DIS-mIoU  |  DIS-mBIoU  | ThinObject-mIoU  |  ThinObject-mBIoU  |
> |------------|:------------:|:------------:|:------------:|:------------:|
> |   SAM (**Baseline**)    |   62.0    |   52.8    |   73.6    |   61.8    |
> |   HQ-SAM (**remove both DIS and ThinObject**)   |   72.9    |   63.1    |   82.7    |   70.7    |
> |   HQ-SAM (**remove DIS**)    |   74.7    |   66.2    |   (90.1)    |   (80.4)    |
> |   HQ-SAM (**remove ThinObject**)    |   (78.4)    |   (70.3)    |   83.3    |   72.1    |
> |   HQ-SAM (**Default HQSeg-44K**)   |   (78.6)    |   (70.4)    |   (89.5)    |   (79.9)    |
>
> The improvement of HQ-SAM over SAM is still substantial on DIS or ThinObject (over 10.0 points on DIS-mIoU and 9.0 points on ThinObject-mIoU), even when the corresponding training splits are removed from training. We will include it in the next revision.
>
> > **Q3.** Experiments without error correction.
>
> Thanks for the suggestion. While we explained error correction in Q1, here we provide results comparison on COCO, LVIS, COIFT and HRSOD before and after using error correction. Error correction improves results consistently on all datasets.
>
> |  Method  |  COCO-AP | COCO-AP^B  |  LVIS-AP  |  LVIS-AP^B  |  COIFT-mIoU  |  COIFT-mBIoU  |  HRSOD-mIoU  |  HRSOD-mBIoU  |
> |------------|:------------:|:------------:|:------------:|:------------:|:------------:|:------------:|:------------:|:------------:|
> |   **Before** error correction   |   48.5    |   33.3    |   43.6    |   38.5    |   92.1    |   86.5    |   90.2    |   83.1    |
> |   **After** error correction    |   49.5    |   34.4   |   43.9    |   38.8   |   94.8    |   90.1    |   93.6    |   86.9    |
>
> > **Q4.** Why selecting these 6 datasets to form HQSeg-44K benchmark?
>
> As in L175 to L177 of paper, to our knowledge, these are six existing image datasets with highly accurate mask labels and extreme boundary details. They provide very accurate manual annotation on objects with complex structures. We show their annotation quality in the attached PDF rebuttal (Fig. 2). In Tab. 7 of the Supp. file, we analyze the effect of replacing the HQ-Seg-44k training data with the same number of masks provided in SA-1B. The results degrade severely, achieving similar performance to the original SAM.
>
> > **Q5.** The speed limitation.
>
> To address the speed limitation, we replace the default Vit-B/L/H with the ViT-tiny backbone provided by MobileSAM, and then follow the HQ-SAM training strategy. The resulting model dubbed as Light HQ-SAM here achieves 45.0 AP on COCO,  reaching a **real-time speed of 41.2 FPS**. Light HQ-SAM only has a 40.3MB model size, but achieves better zero-shot COCO AP than ViT-B based SAM with 358MB model size. We provide a detailed comparison for SAM variants below (also in Fig. 3 of the PDF rebuttal).
>
> |  Model  |  Inference Speed (FPS)  |  Four HQ Datasets mIoU | Four HQ Datasets mBIoU  |  COCO-AP | COCO-AP^B  |  Total #Params(MB) | Trainable #Params(MB)  |   Memory (G)  |
> |------------|:------------:|:------------:|:------------:|:------------:|:------------:|:------------:|:------------:|:------------:|
> |   SAM-B    |  10.1    |  70.6    |   62.3    |   44.4    |   28.2    |   358    |   358    |     5.1    |
> |   HQ-SAM-B    |  9.8    |  86.3    |   78.1    |   46.7    |   31.3    |   362.1    |   4.1    |     5.1    |
> |   SAM-L    | 5.0    |   79.5    |   71.1    |   48.5    |   33.3    |   1191    |   1191    |     7.6    |
> |   HQ-SAM-L    |  4.8    |   89.1    |   81.8    |   49.5    |   34.4    |   1196.1    |   5.1    |    7.6    |
> |   SAM-H    |  3.5    |  75.6    |   68.3    |   48.9    |   34.0    |   2446    |   2446    |     10.3    |
> |   HQ-SAM-H    |  3.4    |   89.3    |   81.5    |   49.9    |   34.9    |   2452.1    |   6.1    |    10.3    |
> |   MobileSAM    |  44.8    |  69.0    |   58.8    |   44.3    |   28.6    |   38.6    |   38.6   |     3.7    |
> |   **Light HQ-SAM**    |**41.2**    |   81.4    |   71.6    |   45.0   |   29.6    |   **40.3**    |   **1.7**    |      3.7    |

---

> > ### Comment · Reviewer_eEa5 · 2023-08-20
> > **Response to authors**
> >
> > Thanks for your great efforts. I decide to raise my score.

---

### Official Review · Reviewer_xJfb · 2023-07-06

**Soundness:** 4 excellent
**Presentation:** 4 excellent
**Contribution:** 4 excellent
**Rating:** 7
**Confidence:** 4

**Summary:**

This paper proposes a high quality version of segments anything. It points out the issues of existing segment anything models in Figure 1 and proposes the corresponding solution. It proposes a high-quality output token and a global-local fusion module. The results are good at various open benchmarks. The visualizations are rich and impressive.

**Strengths:**

1. This paper improves the famous SAM model to a higher grade. This contribution is great and meaningful.

2. The paper itself is also in a high quality. The writing is clear. The idea is direct and easy to understand.

3. The results are impressive. Obvious improvement upon the SAM baseline on several benchmarks show the effectiveness of the proposed method.

4. The proposed dataset is also a good contribution. This makes this area can be followed by more researchers.

**Weaknesses:**

1. The HQ-Output token seems the key point of the proposed method. It should have some distinguished difference to common tokens. Some feature-level visualization should be provided to show these differences. This will make the design explainable.

2. The comparisons in experimental are all among the proposed and SAM. Although this work is designed improve SAM, other related woks should also be compared. For example, the work [33] is also designed for high quality segmentation. Similar to some other related works, they should be compared in at least on dataset.
[33] High quality segmentation for ultra high-resolution images, CVPR 2022

3. The Global-local Fusion model fuse the early and final layers. This design seems very related to the famous feature pyramid network (FPN) in object detection. Did the author try FPN before? FPN fuses more layers and might be a better choice. It would be helpful if the author could provide some ablation on this point.

**Questions:**

Please see the weakness. Overall, I like this work. I will further improve the rate if the weaknesses are well-addressed, but might degrade the rate if not.

**Limitations:**

As stated by the author, speed might be a small concern. My suggestion is that the FPS gap would be smaller if TensorRT is applied. This is not a hard job, as the original SAM already contains a TensorRT conversion tool.

---

> ### Author Rebuttal · Authors · 2023-08-08
>
> We thank the reviewer for acknowledging our paper with great/meaningful contribution, good writing and impressive results. Here we respond to your insightful comments and questions.
>
> > **Q1.** Visualization of HQ-Output Token.
>
> We provide visual comparison of our HQ-Output Token **vs.** SAM's common output token for their cross-attention maps in the last token-to-image layer of the mask decoder. The figure is in the attached one-page PDF rebuttal (Fig. 1). We observe that our HQ-Output Token attends to the boundary and thin structure regions that are missed by the common token. We will include this visual comparison into our final paper.
>
> > **Q2.** Comparison with work CRM [33].
>
> We provide the comparison between SAM, the suggested work CRM [33] and our HQ-SAM below.
>
> |  Method  |  COCO-AP | COCO-AP^B | Four HQ Datasets mIoU | Four HQ Datasets mBIoU  |
> |------------|:------------:|:------------:|:------------:|:------------:|
> |   SAM (baseline)    | 48.5    | 33.3    | 79.5    | 71.1    |
> |   SAM + Cascade PSP [6]    |   13.4    |   2.8    |   80.9    |   74.6    |
> |   SAM + CRM [33]    |   28.7    |   15.9    |   81.4    |   75.4    |
> |   HQ-SAM    | **49.5**    | **34.4**    | **89.1**   | **81.8**    |
>
> Our approach achieves the best performance both on the COCO dataset and the Four HQ Datasets, where CRM [33] shows poor results when generalizing to zero-shot COCO. We will add the comparison to CRM [33] in our paper.
>
> > **Q3.** Ablation experiment on using FPN.
>
> Instead of following FPN to build a feature pyramid for fusion, our Global-local Fusion directly fuses the early-layer and final-layer features by a simple up-sampling and direct element-wise sum up. Here, we provide ablation experiments by replacing Global-local fusion by FPN, where global-local fusion obtains better results both on the COCO and four HQ datasets despite its design simplicity. FPN is designed for ResNet-based backbone, and may not be optimal for ViT based backbone with the same spatial resolution output after each block of the backbone.
>
>
> |  Method  |  COCO-AP | COCO-AP^B |Four HQ Datasets mIoU | Four HQ Datasets mBIoU  |
> |------------|:------------:|:------------:|:------------:|:------------:|
> |   HQ-SAM using **FPN**    |   49.0   |   33.8    |   87.4    |   79.0    |
> |   HQ-SAM using **Global-local Fusion**  | **49.5**    | **34.4**    | **89.1**   | **81.8**    |
>
> We will include this ablation experiment on FPN in our paper.
>
> > **Q4.** The speed limitation.
>
> To address the speed limitation of HQ-SAM, we replace the default Vit-B/L/H with the ViT-tiny backbone provided by MobileSAM [a], and then follow the same HQ-SAM training strategy. The resulting model, which we dub it as Light HQ-SAM, achieves 45.0 AP on COCO attaining a **real-time speed of 41.2 FPS**. Light HQ-SAM only has a 40.3MB model size, but achieves better zero-shot COCO performance than ViT-B based SAM with 358MB model size. Below we provide a detailed comparison among an array of SAM variants.
>
> |  Model  |  Inference Speed (FPS)  |  Four HQ Datasets mIoU | Four HQ Datasets mBIoU  |  COCO-AP | COCO-AP^B  |  Total #Params(MB) | Trainable #Params(MB)  |   Memory  |
> |------------|:------------:|:------------:|:------------:|:------------:|:------------:|:------------:|:------------:|:------------:|
> |   SAM-B    |  10.1    |  70.6    |   62.3    |   44.4    |   28.2    |   **358**    |   **358**    |     5.1G    |
> |   HQ-SAM-B    |  9.8    |  86.3    |   78.1    |   46.7    |   31.3    |   362.1    |   4.1    |     5.1G    |
> |   SAM-L    | 5.0    |   79.5    |   71.1    |   48.5    |   33.3    |   1191    |   1191    |     7.6G    |
> |   HQ-SAM-L    |  4.8    |   89.1    |   81.8    |   49.5    |   34.4    |   1196.1    |   5.1    |    7.6G    |
> |   SAM-H    |  3.5    |  75.6    |   68.3    |   48.9    |   34.0    |   2446    |   2446    |     10.3G    |
> |   HQ-SAM-H    |  3.4    |   89.3    |   81.5    |   49.9    |   34.9    |   2452.1    |   6.1    |    10.3G    |
> |   MobileSAM    |  44.8    |  69.0    |   58.8    |   44.3    |   28.6    |   38.6    |   38.6   |     3.7G    |
> |   **Light HQ-SAM**    |**41.2**    |   81.4    |   71.6    |   45.0   |   29.6    |   **40.3**    |   **1.7**    |      3.7G    |
>
> We provide a comprehensive performance-speed-model size comparison on SAM variants in Figure 3 of the one-page PDF rebuttal. As the reviewer suggests, if TensorRT is further applied, the speed of HQ-SAM and Light HQ-SAM can be further improved. However, even without such optimizations, Light HQ-SAM achieves real-time operating speed.
>
> [a] Zhang, Chaoning, et al. "Faster Segment Anything: Towards Lightweight SAM for Mobile Applications." arXiv preprint arXiv:2306.14289 (2023).

---

> > ### Comment · Reviewer_xJfb · 2023-08-12
> > **Reply to Authors**
> >
> > Many thanks for the detailed reply and experiments. My concerns have all been addressed. I will keep my rate as "7 Accept".

---

### Official Review · Reviewer_2HQS · 2023-07-07

**Soundness:** 3 good
**Presentation:** 3 good
**Contribution:** 3 good
**Rating:** 6
**Confidence:** 4

**Summary:**

This paper proposes HQ-SAM that produces segmentation masks high quality while maintaining SAM's original promptable design, efficiency, and zero-shot generalizability, by only introducing minimal additional parameters. A learnable High-Quality Output Token is injected into SAM's mask decoder. Instead of only applying it on mask-decoder features, the proposed method first fuses them with early and final ViT features for improved mask details. This paper also proposes a dataset of 44K fine-grained masks from several sources. Extensive experiments and comparisons demonstrate the strong performance of the proposed HQ-SAM.

**Strengths:**

-	Improving SAM to produce more accurate masks is an important and also interesting topic, especially with acceptable finetuning cost.
-	The proposed designs make sense and are justified by ablation studies.
-	HQ-SAM is extensively evaluated on 9 diverse segmentation datasets across different downstream tasks, where 7 out of them are evaluated in a zero-shot transfer protocol.


**Weaknesses:**

-	As shown in Table 3, HQ-SAM significantly boosts the performance on Four HQ datasets even if it has consistent settings with SAM (the first two rows). It indicates that training/finetuning on HQSeg-44K is important for HQ-SAM. So what would happen if SAM is also trained on HQSeg-44K dataset.
-	What will the performance be if removing the fusion conv from the last row of Table 3.


**Questions:**

See the weakness part.

**Limitations:**

Yes

---

> ### Author Rebuttal · Authors · 2023-08-08
>
> We thank the reviewer for acknowledging that HQ-SAM is important, interesting and effective with extensive experiments evaluation. Here we respond to your insightful comments and questions.
>
> > **Q1.** What would happen if SAM is also trained on HQSeg-44K dataset?
>
> We provided the experiment of training SAM on HQSeg-44K in Table 4 of the paper, where we finetuned the whole mask decoder of SAM (4th row) or finetuned the output token of SAM (5th row). We find that while it will bring improvement over SAM on the four HQ datasets, it still achieves inferior performance than HQ-SAM on zero-shot COCO. Here, we also provide an experiment comparison by training the whole SAM on HQ-Seg44K as below:
>
> | Method | COCO-AP | COCO-AP^B |Four HQ Datasets mIoU | Four HQ Datasets mBIoU |
> |----------|:----------:|:----------:|:----------:|:----------:|
> | SAM (baseline)    | 48.5    | 33.3    | 79.5    | 71.1    |
> | Training the whole SAM    | 5.5    | 0.2    | 38.0    | 12.2    |
> | Training SAM's mask decoder   | 19.5    | 9.0    | 87.6    | 79.5    |
> | Training SAM's output token    | 48.7    | 33.7    | 87.6    | 79.7    |
> | **HQ-SAM**    | **49.5**    | **34.4**    | **89.1**   | **81.8**    |
>
>
> Our approach achieves the best results on both the four HQ datasets and COCO. We will include this analysis in the next revision of the paper.
>
> > **Q2.** Experiment on removing fusion conv from the last row of Table 3?
>
> We provide the experiment on removing fusion conv from the last row of Table 3 below:
>
> |  Method  |  Fusion Conv  |  Mask Decoder Feature  |  Final-layer of ViT encoder  |  Early-layer of ViT encoder  |  Four HQ Datasets mIoU  | Four HQ Datasets mBIoU |
> |------------|:------------:|:------------:|:------------:|:------------:|:------------:|:------------:|
> |   HQ-SAM    |     |   &check;    |   &check;   |   &check;    |   88.6   |   81.1    |
> |   HQ-SAM    |   &check;   |   &check;   |   &check;    |  &check;    |   **89.1**   |   **81.8**    |
>
> Also in this case, we find that the fusion conv benefits segmentation performance. We will update the Table 3 of paper with the above results.

---

> > ### Comment · Reviewer_2HQS · 2023-08-21
> >
> > Thanks for the response. Most of my initial concerns have been addressed and I tend to keep my original rating.

---

### Official Review · Reviewer_E7oL · 2023-07-08

**Soundness:** 3 good
**Presentation:** 3 good
**Contribution:** 3 good
**Rating:** 5
**Confidence:** 5

**Summary:**

This paper proposes HQ-SAM, an improved SAM with higher-quality segmentation performance. HQ-SAM freezes the entire SAM model and appends additional learnable modules concurrently to SAM. With the HQ-output token and global-local feature fusion, HQ-SAM achieves better segmentation results and only costs marginal extra resources.

**Strengths:**

1. Additional training cost of HQ-SAM is marginal compared to SAM, but contributes to good performance boost.

2. The idea of adding a new HQ-token specialized for HQ mask is interesting and reasonable to me.

**Weaknesses:**

1. Although the author claim the necessity of freezing the entire SAM model, it's better to verify this by ablation study, for example, unfreezing the entire SAM, or using PEFT methods (LoRA, adapters) with performance-parameter comparison.

2. I'm curious about how HQ-SAM performs in PerSAM (personalization segmentation by SAM) ?

Personalize segment anything model with one shot (https://arxiv.org/pdf/2305.03048.pdf)

**Questions:**

None

**Limitations:**

Yes

---

> ### Author Rebuttal · Authors · 2023-08-08
>
> Thank you for acknowledging our work with an interesting and reasonable design, good performance and marginal training cost. Here we respond to your insightful comments and questions.
>
> > **Q1.** Experiments on unfreezing the entire SAM, or using PEFT methods (LoRA, adapters) with performance-parameter comparison.
>
> Please note that we already provided the experiment on unfreezing SAM’s whole mask decoder or SAM’s output token in Table 4 of the paper. We also studied using PEFT methods as adapter in Table 2 of the Supp. file. Combined with these experiment trials, we further provide a comprehensive zero-shot performance-parameter comparison by also unfreezing the entire SAM and using LoRA method as below:
>
> | Method | COCO-AP | COCO-AP^B | Total #Params(MB) | Trainable #Params(MB)|
> |----------|:----------:|:----------:|:----------:|:----------:|
> | SAM (**baseline**)    | 48.5    | 33.3    | 1191    | -    |
> | **Unfreeze the entire SAM**    | 5.5   | 0.2    | 1191    | 1191   |
> | Unfreeze SAM's mask decoder   | 19.5    | 9.0    | 1191    | 20.6    |
> | Unfreeze SAM's output token    | 48.7    | 33.7    | 1191    | 0.5    |
> | SAM + **LoRA**    | 43.7    | 28.6    | 1192.5   | 1.5    |
> | SAM + Adapter    | 44.8   | 29.6    | 1203   | 12.0   |
> | **HQ-SAM**    | **49.5**    | **34.4**    | 1196.1    | 5.1    |
>
> Among these SAM training strategies with the same training setting, HQ-SAM achieves the best zero-shot COCO performance with marginal training cost. We will add this detailed comparison in the next revision.
>
> > **Q2.** How HQ-SAM performs in PerSAM?
>
> Thank you for the suggestion. PerSAM is a concurrent arXiv work released just several days before the abstract submission deadline. PerSAM focuses on one-shot segmentation, while HQ-SAM is designed for zero-shot high-quality segmentation. Per the reviewer's suggestion, we here analyze the impact of using our HQ-SAM in PerSAM. Our HQ-SAM improves the PerSAM performance on DAVIS 2017 (Table 2 of the PerSAM paper) as shown below:
>
>
> |  PerSAM Variant  |  Backbone  |  J&F  |  J  |  F  |
> |------------|------------|------------|------------|------------|
> |   PerSAM-F     |   ViT-Huge    |   71.85    |   68.98    |   74.72    |
> |   PerSAM-F + **HQ-SAM**    |   ViT-Huge    |   **72.74**    |   **70.00**    |  **75.49**    |
>
>
> Furthermore, equipped with the same Grounding-DINO [a] detector as box prompt, we also performed experiments by replacing SAM with HQ-SAM in Grounded-SAM, and obtained the **first place** in the Segmentation in the Wild (SGinW) competition on the zero-shot track (hosted in CVPR 2023 workshop). Note that SGinW benchmark consists of **25** zero-shot in-the-wild segmentation datasets for evaluation, and Grounded-HQ-SAM outperforms Grounded-SAM obviously using the same detector. The benchmark leaderboard result is publicly available.
>
> [a] Liu, Shilong, et al. "Grounding dino: Marrying dino with grounded pre-training for open-set object detection." arXiv preprint arXiv:2303.05499 (2023).

---

### Author Rebuttal · Authors · 2023-08-08

We appreciate the positive and insightful comments from all four reviewers. We are glad that they found our paper "well motivated and interesting” (E7oL, 2HQS, eEa5), “important and a great contribution” (2HQS, xJfb),  “effective with extensive experiments” (E7oL, 2HQS, xJfb),  “with marginal training cost” (E7oL, 2HQS), "achieving impressive results" (xJfb), and “well written” (xJfb, eEa5).

HQ-SAM is the first work to upgrade SAM for **high-quality** segmentation, achieving consistent performance gains on a wide range of zero-shot segmentation benchmarks with marginal cost. HQ-SAM offers timely insights into how to leverage and extend SAM-like foundational segmentation models in a data-efficient and computation-affordable manner to achieve high-quality results. Both our code and dataset are public to facilitate future research. We address all questions in separate responses to each reviewer, and will incorporate all valuable feedbacks into the our paper.

---

### Author Response · Authors · 2023-08-18
**Sincere Request for Further Feedback**

Dear Reviewers,

Firstly, we would like to express our thanks for the time and effort you’ve spent in reviewing HQ-SAM. As we approach the end of the discussion period, we eagerly anticipate your feedback on our recent response. We’ve endeavored to incorporate the suggested experiments and elaborate on our methods based on your insightful comments. Should there be any ambiguities or further questions, please know that we are more than willing to provide clarity or delve deeper into any topic. Your continued guidance is really appreciated.

Warm regards,

The Authors

---

### Decision · Program_Chairs · 2023-09-21

**Decision:**

Accept (poster)

**Comment:**

This paper proposes HQ-SAM, which is an improved SAM with higher-quality segmentation accuracy. HQ-SAM freezes the SAM model and appends additional learnable modules to SAM. With the HQ-output token and global-local feature fusion, HQ-SAM achieves better segmentation results and only costs marginal extra resources.

All reviewers are positive about the paper after the rebuttal. The authors should also include the additional results presented in the rebuttal to reviewers E7oL, eEa5, xJfb in the final version.